# Prediction of 3D Cardiovascular hemodynamics before and after coronary artery bypass surgery via deep learning

Gaoyang Li[1], Haoran Wang[1,2], Mingzi Zhang [1], Simon Tupin [1], Aike Qiao[3], Youjun Liu[3], Makoto Ohta[1,2,4] & Hitomi Anzai [1✉]

The clinical treatment planning of coronary heart disease requires hemodynamic parameters to provide proper guidance. Computational fluid dynamics (CFD) is gradually used in the simulation of cardiovascular hemodynamics. However, for the patient-specific model, the complex operation and high computational cost of CFD hinder its clinical application. To deal with these problems, we develop cardiovascular hemodynamic point datasets and a dual sampling channel deep learning network, which can analyze and reproduce the relationship between the cardiovascular geometry and internal hemodynamics. The statistical analysis shows that the hemodynamic prediction results of deep learning are in agreement with the conventional CFD method, but the calculation time is reduced 600-fold. In terms of over 2 million nodes, prediction accuracy of around 90%, computational efficiency to predict cardiovascular hemodynamics within 1 second, and universality for evaluating complex arterial system, our deep learning method can meet the needs of most situations.

[1] Institute of Fluid Science, Tohoku University, 2-1-1, Katahira, Aoba-ku, Sendai, Miyagi 980-8577, Japan. [2] Graduate School of Biomedical Engineering, Tohoku University, 6-6 Aramaki-aza-aoba, Aoba-ku, Sendai, Miyagi 980-8579, Japan. [3] College of Life Science and Bioengineering, Beijing University of Technology, No.100, Pingleyuan, Chaoyang District, Beijing 100022, China. [4] ELyTMaX UMI 3757, CNRS–Université de Lyon–Tohoku University, Sendai, Miyagi 980-8579, Japan. ✉email: hitomi.anzai.b5@tohoku.ac.jp

Coronary heart disease (CHD) has become a leading cause of global mortality[1,2], with more than 50% of these cases being related to coronary stenosis[3]. In order to achieve successful therapeutic effects, CHD clinical treatment plans require a variety of hemodynamic parameters to provide proper guidance. Currently, pressure field-based fractional flow reserve (FFR) is the gold standard for the clinical diagnosis of myocardial ischemia severity caused by stenosis[4–7]. The treatment regimen guided by FFR has been proven safe and effective[8,9]. For patients with severe myocardial ischemia, revascularization is typically required. Coronary artery bypass grafting (CABG) is the most commonly used revascularization procedure[10]. Velocity field-based postoperative blood flow changes in the grafts and the distal end of the stenotic coronary artery are thought to be the most direct parameter for assessing the influence of CABG[11,12]. However, the application rate of hemodynamic parameters in clinical practice is low, mainly due to its high measurement cost and potential risks during catheter insertion. Taking invasive FFR as an example, the price of the pressure guide wire required for measurement is relatively high. In addition, the use of vasodilator drugs such as adenosine may increase the time and cost of interventional surgery[13], and may also increase the surgical risk of patients with adenosine sensitivity or asthma[14]. Therefore, obtaining hemodynamic parameters, including velocity and pressure inexpensively and non-invasively is crucial for the support of CABG and the treatment of CHD.

A large number of previous studies have used computational fluid dynamics (CFD) to obtain cardiovascular hemodynamics[15–19]. Based on the patient's cardiovascular geometry, provided by medical imaging data (e.g., MRI, CT, etc.) and given boundary conditions, CFD can inexpensively and non-invasively obtain solutions for velocity and pressure through the conservation of mass and momentum under isothermal and incompressible assumptions. However, the cost to model cardiovascular hemodynamics with available computational resources is very high[20]. When subjects' personalized CFD boundary conditions (e.g., the inlet is set to pulsatile flow, and the outlet pressure is an invasive measured value) are used to calculate the hemodynamics of complex cardiovascular models with small grafts and coronary branches, even high-performance computational clusters usually require several hours of iteration to ensure the accuracy of the model. Even with simplified boundary conditions (e.g., steady flow at the inlet and zero pressure at the outlet), CFD methods require a calculation time around 10 min. In addition, each patient's cardiovascular geometry is unique. This means that the CFD procedure will need to be completed separately and repeatedly for each patient. The high computational cost of CFD hinders its clinical application to the treatment of CHD, such as the inability to provide surgical guidance. Therefore, it has become increasingly necessary to develop a cardiovascular hemodynamic calculation method that can reduce calculation costs while ensuring model accuracy.

The development of deep learning, one of many machine learning methods, provides a new way to solve the above problems. Deep learning detects distributed representation features of data by constructing neural networks and combining low-level features to form more abstract high-level features or attribute features, thus completing the task of classification or regression[21–27]. Advanced deep learning algorithms and high-performance GPU servers can greatly reduce computing times while ensuring high accuracy. Due to the development of deep learning techniques, some studies have introduced its application to predict 2D/3D flow fields from geometrical features. For example, Guo et al. put forward a calculation method of 2D flow around simple geometric models based on convolutional neural networks[28]. And Liang et al. proposed a deep learning method to predict 3D simplified thoracic aortic hemodynamics[29]. However, the research concerning predictions of hemodynamics via deep learning is still very limited[30]. The main limitations of these studies are: (1) most studies focus on 2D flow fields, which have a limited scope of application[31–34], (2) the 3D flow field model only appears in ideal geometry, and sample resolution in the dataset is too low to represent complex flow field distributions and geometric structures[28,29]. For CABG surgery, a cardiovascular model with small grafts and coronary branches has an intricate geometry and internal flow field distribution. Therefore, in this study, in order to accurately predict complex 3D cardiovascular hemodynamics with limited samples, new requirements to adapt to the flexibility and high resolution of the input geometry have been imposed on datasets and deep learning networks, which is also the main technical problem and contribution of this study. Concerning the dataset, each sample must have enough spatial resolution to resolve complex flow field and model geometry. Therefore, it is necessary to find a new, high-resolution sample representation format. And concerning the network, it is necessary to propose a new network structure that can effectively handle the new sample format.

In this study, with the aim of predicting 3D hemodynamics in the real cardiovascular systems of patients with coronary stenosis (e.g., geometry containing aorta, coronary arteries, and bypass graft), we have proposed a new deep learning method that could predict the velocity field and pressure field based on the geometric features of the model. We collected cardiovascular data with small branches from computed tomography angiography (CTA) performed on 110 patients with CHD for model expansion and simulation of CABG surgery. Under certain boundary conditions, a CFD method was used to obtain the hemodynamic results (i.e., velocity and pressure field) of all models. Later, we converted the CFD results into high-density 3D point clouds. The point cloud inherited the ability of CFD results to characterize the geometric structure and flow field distribution of the model, which could characterize the complex flow field distribution and geometry of real cardiovascular models with high resolution[35,36]. On this basis, preoperative and postoperative cardiovascular hemodynamic point datasets were established, respectively. We also proposed a new deep learning network based on the PointNet structure[37], which could effectively resolve the disorder of point clouds and introduce spatial relationships. By extracting and integrating global and local features of the point cloud, the network could analyze and reproduce the relationship between vessel geometry in the point cloud datasets and the corresponding hemodynamics. The deep learning network only needs to be trained once. Next, when we input cardiovascular geometry information from a new patient, the corresponding 3D hemodynamic parameter prediction results could be obtained within 1 second. In order to verify the accuracy of our deep learning method, we define error functions (ERR), normalized mean absolute error (NMAE), and mean relative error (MRE) to evaluate the difference between the two methods. Based on the acquired hemodynamic results, we further calculated and compared the preoperative FFR and the postoperative blood flow of the graft and the distal stenotic coronary artery. Statistical analysis results showed that the predicted results of deep learning were in agreement with the traditional CFD method, but the calculation time was reduced 600-fold. Our deep learning method aims to realize the prediction of velocity and pressure fields before and after CABG surgery instead of CFD. To the best of our knowledge, this study represents the first report describing deep learning techniques that can effectively and accurately predict 3D hemodynamics of complex cardiovascular system with small grafts and coronary branches with limited data.

## Results

**Prediction results of velocity field**. When the preoperative and postoperative velocity field datasets were used as inputs to the proposed deep learning network, the loss function value versus epochs was made available (as seen in Supplementary Fig. 1). The loss function fully converged.

We compared the prediction results of deep learning with CFD. Figure 1 displays the streamline diagram of a 3D velocity field distribution and a cross-sectional view of velocity magnitude contour in the same areas. It showed that the velocity fields obtained by the two methods had good reliability. Our deep learning method could predict the distribution of velocity fields in the entire cardiovascular model before and after CABG, which included not only general attributes such as laminar blood flow, but also the occurrence of complex vortex structures.

We calculated the mean value of the predicted velocity field ERR of the 100 models in the test sets, as shown in Table 1. The result showed that the prediction accuracy of the coronary arteries (NMAE < 3%, MRE < 5%) and grafts (NMAE < 1.5%, MRE < 2.5%) was higher than that of the aorta and superior aortic branch artery (Preoperative: NMAE = 6.02%, MRE = 9.77%; Postoperative: NMAE = 6.01%, MRE = 9.74%). This was mainly due to the larger magnitude of flow and the complicated flow field distribution by vortex and flow separation in both the aorta and superior aortic branch artery parts. We give detailed explanations and feasible improvements in the prediction error analysis section below.

Deep learning prediction results (shown in Fig. 1 and Table 1) could reflect the effect of CABG on the distribution of internal flow field in the cardiovascular system. It could accurately reproduce velocity fields in the small lesion coronary and reconstructed grafts, which meant that it could not only reflect the preoperative ischemic condition of LAD branches with different stenosis rates, but also signal the postoperative improvement of blood supply. In addition, it could be seen from the predicted results that CABG had a big impact on the flow field of the grafts and LAD with stenosis but had little influence on the flow field of other parts. The proposed network could effectively identify significant and non-significant disturbances of the graft on the flow field, which highlighted its high performance.

**Prediction results of pressure field**. This study aimed to develop a universal deep learning method. The same network structure could accomplish predicting hemodynamic parameters with different attributes, which could then be proven via the analysis results of the velocity field and pressure field.

Different from the velocity, the pressure in the flow field was scalar, that was, the pressure at a point had the same value in all directions. There were different vector components of velocity vector in X, Y, and Z directions. Therefore, the pressure datasets as the network input contained less information than the velocity field datasets, which was reflected in the convergence speed of the loss function value versus epochs (as seen in Supplementary Fig. 1). The loss function converged faster.

Figure 2 displays a 3D pressure distribution cloud map obtained from deep learning and CFD method, with a cross-sectional view of the same part. The pressure fields obtained by the two methods were also in agreement. Our deep learning method could accurately replicate the pressure distribution of the entire cardiovascular model with small grafts and coronary branches.

We calculated the mean value of the predicted pressure field ERR of the 100 models in the test set, as shown in Table 2. The prediction accuracy of pressure fields of coronary artery (NMAE < 2.5%, MRE < 4%) and grafts (NMAE < 1.5%, MRE < 2%) was

also higher than that of the aorta and superior aortic branch artery (Preoperative: NMAE = 4.30%, MRE = 7.61%; Postoperative: NMAE = 4.28%, MRE = 7.35%), as explained in the prediction error analysis section below.

Based on the velocity and pressure field, we calculated important clinical indicators related to CABG surgery: preoperative FFR of the lesion LAD and the postoperative blood flow of the graft and the distal stenotic coronary artery. The performance of our deep learning method could be further evaluated by comparing the indicators obtained by the two methods, which is described in detail below.

**Deep learning improves computing efficiency**. After the training was completed, and when the point coordinate space information of the cardiovascular model in the test set was input to the network, the hemodynamics of the query point could be obtained within 1 second using a NVidia GeForce GTX 1080 Ti GPU. For the CFD method, the calculation time of one model on an Intel Xeon Gold 6148 2.4 Ghz × 2 CPU server was about 10 min. Deep learning improved the computational efficiency of a single model 600-fold. Although it took some time (about 40 h) to complete network training, this process only needs to be completed once to predict the hemodynamic of all models in the given test set. Compared to the traditional CFD method, where each model requires independent simulation calculations, the computational cost of deep learning is far less than CFD.

Together with the accuracy analysis of the results, the proposed deep learning network could efficiently and accurately predict 3D hemodynamics of complex cardiovascular system with small grafts and coronary branches. This also meant that deep learning has broad application prospects, such as the possibility of application in the early planning or even real-time support of CABG.

**Calculate FFR and improved flow based on prediction results**. In order to further confirm the accuracy and clinical utility of our deep learning method, we calculated the preoperative FFR value and postoperative blood flow value of the grafts and the distal end of the stenosis using the hemodynamic results acquired from CFD and deep learning, respectively. The FFR was defined as the ratio of the mean pressure at a cross-section 3 cm downstream of the stenosis to the mean pressure at the LAD coronary entry region[38]. Improved blood flow was defined as graft inlet flow, which was calculated based on the diameter and velocity of the graft inlet section. Figure 3a, b were the scatter plots of FFR and improved flow on each model obtained from the two methods, which showed that the correlation between the FFR and improved flow attained from these methods was excellent (FFR: $r = 0.9580$, $P < 0.001$; Flow: $r = 0.9734$, $P < 0.001$). Also, the Bland–Altman analysis result is as shown in Fig. 3c, d: 97 sets of FFR data and 97 sets of improved flow data fall within the 95% confidence interval (FFR: $-0.07780–0.09254$; flow: $-1.282–0.8568$), which confirmed that the clinical indicators calculated by these methods were in agreement.

Our deep learning method reduced the computational time of hemodynamics to 1 s, and its output was a point cloud format which was easy to post-process. On this basis, the calculation time of clinical indicators such as FFR and improvement of blood flow could also be reduced to a few seconds while ensuring high accuracy, which further confirmed the superiority of our deep learning method.

**Prediction error analysis**. We extracted regions with large prediction error function values (MRE > 10%) in the entire cardiovascular model. These regions were highly consistent with the

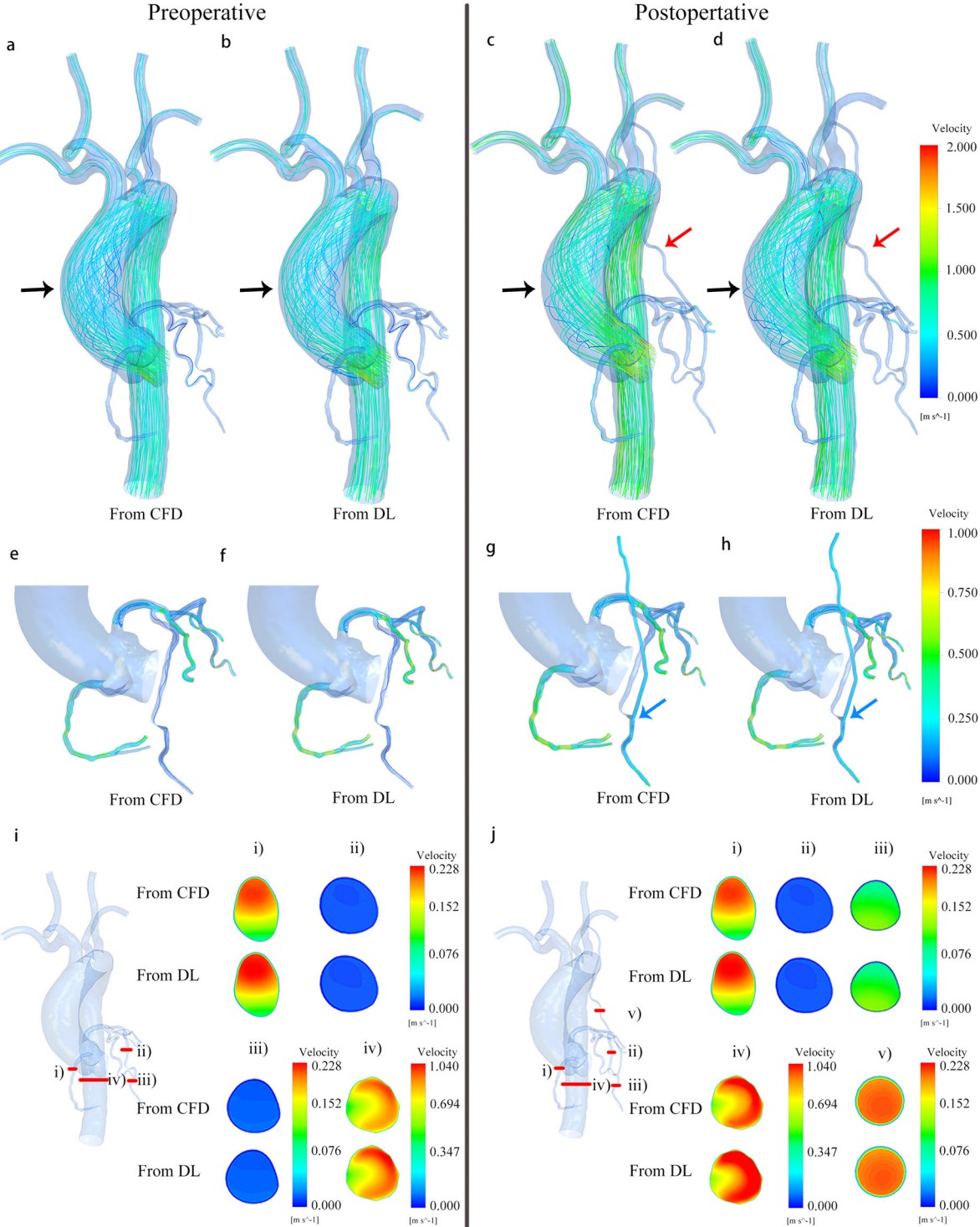

**Fig. 1 Comparison of velocity fields from CFD and deep learning (DL) methods, using a sample with a stenosis rate of 85%. a–d** Are streamline diagrams of the entire cardiovascular internal flow field before and after CABG obtained from CFD and DL. The position indicated by the black arrow in **a–d** is the vortex region of ascending aorta. The position indicated by the red arrow in **c**, **d** is the grafts. **e–h** Are streamline diagrams of the coronary artery and grafts flow field. The blood flow velocity inside the coronary artery is smaller than that in the aorta, which means that it is difficult to clearly show the distribution of the internal flow field in the coronary artery under the same velocity color bar. Therefore, the coronary flow field is displayed separately. The position indicated by the blue arrow in **g–h** is the connection site between the graft and the LAD. **i**, **j** Are cross-sectional views of the velocity distribution, respectively from (i): RA branch; (ii): the proximal end of LAD before stenosis; (iii): the distal end of LAD after stenosis; (iv): descending aorta; (v): graft. (v) can reflect laminar flow in the graft. The comparison results confirm the high consistency of the velocity field obtained by the two methods. This clearly shows the effect of CABG surgery on the flow field distribution of the entire cardiovascular system.

**Table 1 Performance evaluation of the velocity field.**

| | ERR(%) | Proximal end of LAD | Distal end of LAD | LCX | RA | Graft | Aorta and superior aortic branch artery |
|---|---|---|---|---|---|---|---|
| Preoperative | NMAE | 2.62 ± 1.47 | 2.53 ± 1.02 | 2.33 ± 1.25 | 2.91 ± 1.64 | | 6.02 ± 2.97 |
| | MRE | 4.12 ± 2.46 | 3.97 ± 2.77 | 4.61 ± 2.13 | 4.35 ± 1.87 | | 9.77 ± 3.86 |
| Postoperative | NMAE | 2.60 ± 1.43 | 2.64 ± 1.25 | 2.33 ± 1.25 | 2.91 ± 1.64 | 1.12 ± 0.57 | 6.01 ± 2.96 |
| | MRE | 4.11 ± 2.42 | 4.21 ± 2.96 | 4.61 ± 2.13 | 4.35 ± 1.87 | 2.01 ± 1.25 | 9.74 ± 3.83 |

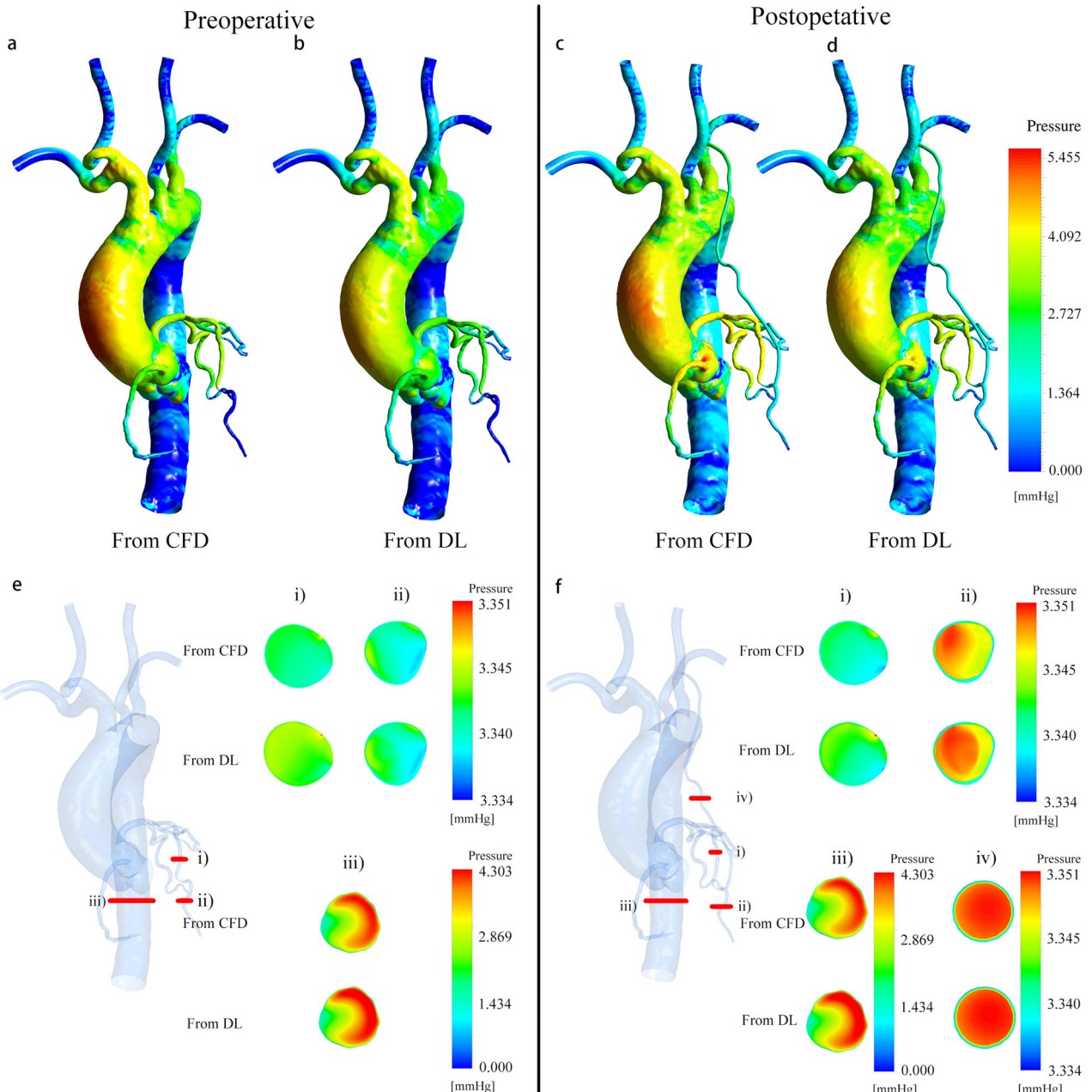

**Fig. 2 Comparison of pressure field from CFD and deep learning (DL), using a sample with a stenosis rate of 85%.** Because the CFD outlet boundary was set as zero pressure condition, the pressure value in this figure was actually the pressure difference relative to the coronary outlet. **a–d** Are pressure distribution cloud maps of the entire cardiovascular before and after CABG obtained from CFD and DL. **e, f** Are cross-sectional views of the pressure distribution, respectively from (i): the proximal end of LAD before stenosis; (ii): the distal end of LAD after stenosis; (iii): descending aorta; iv): graft. The comparison results confirm the high consistency of the pressure field obtained by the two methods. The results of coronary pressure prediction can help us calculate FFR to further evaluate the performance of our deep learning method.

**Table 2 Performance evaluation of the pressure field.**

|  | ERR(%) | Proximal end of LAD | Distal end of LAD | LCX | RA | Graft | Aorta and superior aortic branch artery |
|---|---|---|---|---|---|---|---|
| Preoperative | NMAE | 2.03 ± 1.13 | 1.83 ± 1.18 | 1.71 ± 1.49 | 2.04 ± 1.12 |  | 4.30 ± 1.58 |
|  | MRE | 3.55 ± 1.74 | 3.12 ± 1.63 | 3.52 ± 1.97 | 3.63 ± 1.96 |  | 7.61 ± 1.99 |
| Postoperative | NMAE | 1.99 ± 1.09 | 1.96 ± 1.31 | 1.71 ± 1.49 | 2.04 ± 1.12 | 1.01 ± 0.47 | 4.28 ± 1.55 |
|  | MRE | 3.52 ± 1.72 | 3.57 ± 1.98 | 3.52 ± 1.97 | 3.63 ± 1.96 | 1.98 ± 0.97 | 7.35 ± 1.89 |

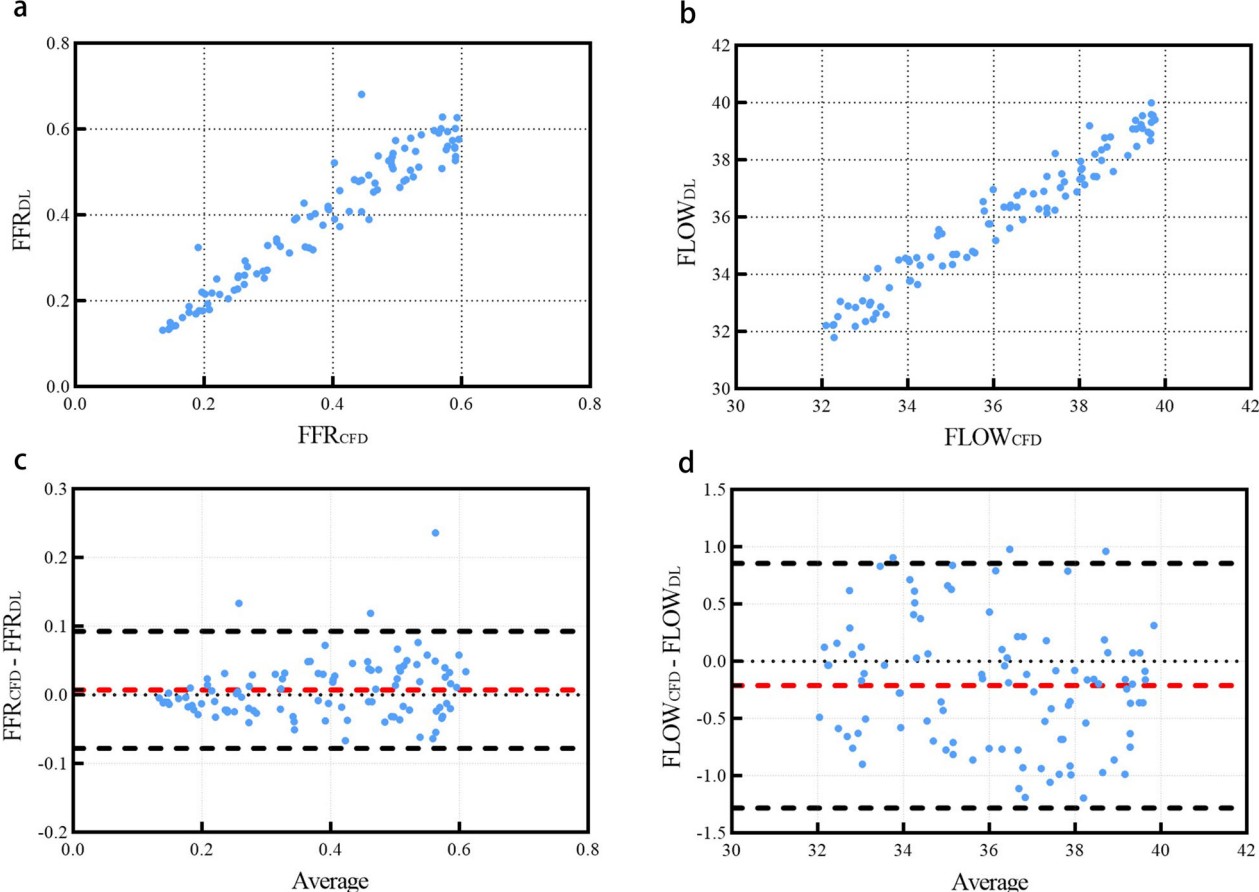

**Fig. 3 Comparison of FFR and improved blood flow obtained from CFD and deep learning. a** Is a scatterplot of FFR values from CFD (FFR$_{CFD}$) and deep learning (FFR$_{DL}$). **b** Is a scatterplot of improved blood flow values from CFD (FLOW$_{CFD}$) and deep learning (FLOW$_{DL}$). **c, d** Are Bland–Altman analysis plot of corresponding (**a, b**). The hemodynamic results used to calculate FFR and improved blood flow are from 100 cardiovascular models in the test set.

vortex regions in the CFD calculation results, as shown in Fig. 4a, b. Vortexes were mainly distributed in the aorta and superior aortic branch artery rather than the coronary artery and graft. Points in the aorta and superior aortic branch artery region accounted for more than 99% of the query point cloud, and more than 30% of the points in the whole region were located in the vortex region, which was the main source of prediction errors for the cardiovascular model. We extracted the points only in the vortex region which was defined with Eigen Helicity method, level 0.005, actual value 44.89 s$^{-1}$ for predicted results and calculated the error as shown in Fig. 4c. The points in the coronary artery and graft part only accounted for 1% of the query point cloud, and only about 10% of them were in the vortex region, which had little effect on the prediction errors. The vortex distribution also explained why the graft and coronary parts had higher prediction accuracy.

Compared to laminar flow, the vortex flow part was extremely complicated. Previous studies that used deep learning to predict complex vortexes required much more training data than ours, even in 2D space[39–41]. Taking Lee's research as an example[39], in a 2D plane with a size of 250 × 250 (grid cells), 500,000 vortex samples were needed to train the network. The number of samples was far more than that of this study. However, the complexity of the vortex (2D) was lower than that of this study (3D). Therefore, we hypothesized that the limited dataset of this study was not sufficient to fully characterize the characteristics of vortex, which could lead to a decrease in the accuracy of the prediction results. To verify this theory, we fixed the test set and increased the size of the training set from 10% to 100% and then calculate the MRE of the vortex region, as shown in Fig. 4c. Even at the maximum of the training set, the MRE of the vortex part still had a downward trend. With a larger dataset, accuracy would also improve. The collection of more cardiovascular models of patients with coronary artery stenosis to build a richer dataset is necessary as more data will lead to higher prediction accuracy and better model performance.

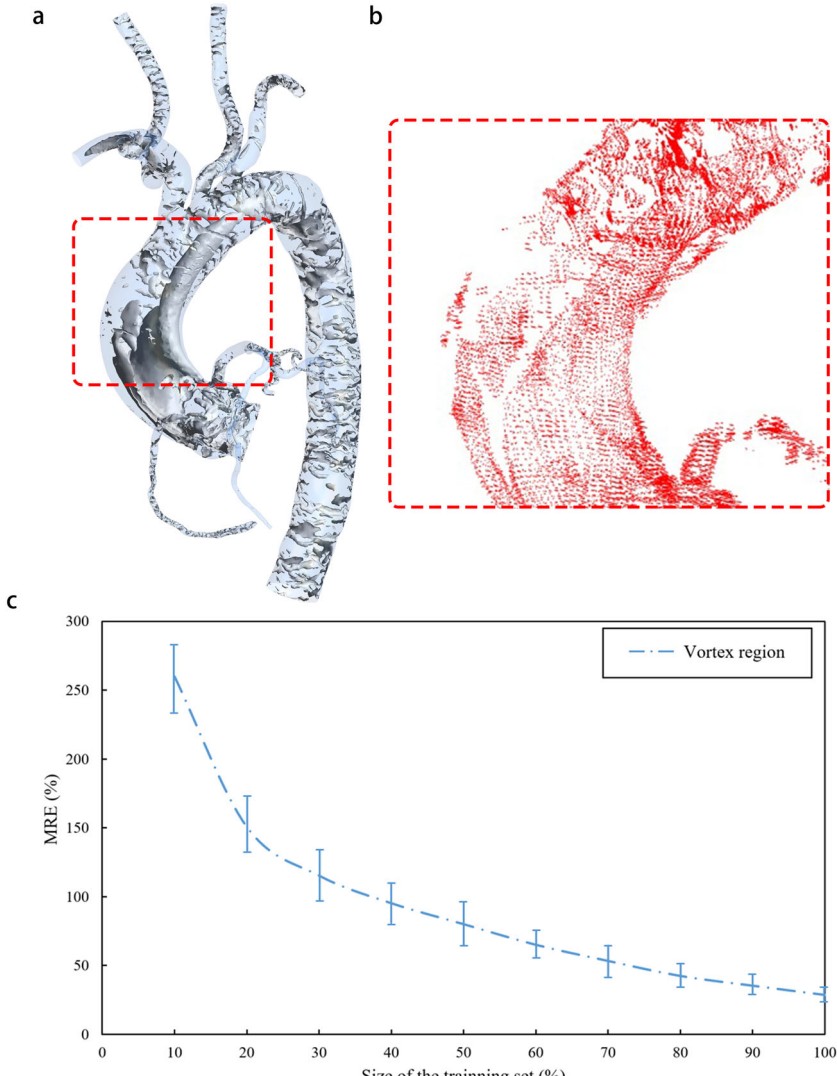

**Fig. 4 Prediction error analysis.** Using a model in the preoperative test set as a sample, **a** shows the vortex regions in its CFD simulation flow field, which are mostly distributed in the aorta and superior aortic branch artery rather than the grafts and coronary arteries. Compared to the entire model, the number of points in the vortex region of grafts and coronary arteries only accounted for 0.1% of all query points. For most models, there is an obvious vortex region in the ascending aorta area circled by the red box. In order to clearly indicate the distribution of points with high prediction errors (MRE > 10%), **b** shows the points with high errors in deep learning predicted velocity field of the same model as (**a**) (the circled area). The distribution of these points is highly consistent with the vortex region in (**a**), which proves that the error mainly comes from the vortex region of the aorta and superior aortic branch artery. **c** Shows the prediction errors along to the different size of training set only in the vortex region. **c** Shows the influence of the training set size on MRE of the vortex region. We fix the test set and increase the size of the training set from 10% to 100%. Then we calculate and observe the MRE of the vortex region. When the size of the training set reach 100%, MRE still displays a downward trend. The minimum value of 28.53% is still far greater than the aorta and superior aortic branch artery MRE of 9.74 ± 3.83% as shown in Table 1. It further confirms the source of the prediction error and shows that it is necessary to increase the size of training sets.

## Discussion

In this study, we used deep learning to predict 3D hemodynamics of complex cardiovascular systems with small grafts and coronary branches before and after CABG surgery. Our results showed that calculation results of the deep learning and CFD methods were highly consistent, and the calculation efficiency was improved 600-fold. This study proved that deep learning could achieve efficient and accurate predictions of 3D hemodynamics in complex models, which also means that it has great application value in scientific research and clinical fields.

The comparative analysis of our deep learning method against previous deep learning approaches is available in Supplementary Table 1. As mentioned above, there was no previous research that achieved 3D hemodynamic prediction of complex models such as

thoracic aortic tree, including the small grafts and coronary branches. Our work made up for this gap, which was mainly due to the use of the point cloud to create datasets and propose a high-performance deep learning network. Previous studies typically required normalized flow field data to help deep learning networks obtain correspondence between model geometry and flow field distribution[24,25]. In other words, regular spatial relationships (e.g., orthogonalization) are introduced into the flow field data by interpolation or other methods. Therefore, the spatial resolution of flow field data depends on space segmentation size during normalization. When there are big differences in the size of different parts of the model, such as the radius of the aorta which is much larger than that of the coronary artery and the graft in this study, it will be very difficult to choose the

appropriate space segmentation size. Large segmentation sizes cannot characterize small parts of the model. Small segmentation size results in a large amount of data exceeded the upper limit of server processing. For example, when using Guo et al.'s normalization method to process the cardiovascular models in this study[28], we should place the model in a three-dimensional space cuboid. The length, width, and height of the cuboid should be the maximum value in the corresponding direction of the model. For a cardiovascular model with a length of 80 mm, a width of 120 mm and a height of 95 mm, when using 0.1 mm as the segmentation size, one 3D cardiovascular model needs to have $800 \times 1200 \times 950 = 836,000,000$ normalized points to contain the mean shape, which is unacceptable for network and GPU processors. Liang et al. proposed a normalization method for deep learning[29]. However, it was only suitable for large ideal aorta, not for the whole complex cardiovascular system, especially for small coronary branches. In addition, Liang et al. normalized the thoracic aortic models of different patients to the same meshes with only 80,100 nodes. However, in this study, the mesh independence test (as seen in Supplementary Fig. 4) showed that for the thoracic aortic, when the number of meshes exceeded one million, the CFD simulation results can be considered to be stable. Therefore, the deep learning method proposed by Liang et al. still has great limitations in the application range and data resolution. Due to limitations of resolution and network performance, most of the previous research objects are simple idealized flow fields. Unlike previous approaches, we utilized high-density 3D point clouds to build datasets. A point cloud is the connection point of CFD meshes and is generally called a node. It is directly output by the mesh result and does not require normalization or other processing. The point cloud can represent complex or small features of the model under the appropriate mesh setting[42,43]. In this study, the mesh independence test results confirmed that the model contains about 2 million points that could represent the complex structure of the entire cardiovascular system, of which 0.4 million were surface model point clouds and 1.6 million were internal query point clouds. Each point has only spatial coordinates and hemodynamic information, which means there is no connection or interaction between them. Thus, it can store a lot of useful information with a small amount of data unlike a connected point set. In order to resolve the disorder of the point cloud and introduce the spatial relationship, we propose a dual sampling channel network structure based on PointNet. Since there is no connectivity information between the points, there is no specific input sequence for the points. That is to say, when N points are used as input to the network, due to the different input sequence, there may be N! input permutations, that is, the disorder of the point cloud. Symmetric function can ensure that the output is the same regardless of the order of input, to resolve point disorder[37]. The dual sampling channel can extract the geometry of the model point cloud of the surface as the global geometry feature, and the internal query point cloud distribution as the local flow field feature. Global features convey the outer geometry information within the model, which can help the query point cloud to obtain its position inside the model. Under the uniform CFD boundary conditions, the position of the query point is corresponding to the flow field. The local flow field characteristics and the corresponding position information can be used as teacher signals to help the network learn the hemodynamic values of a specific position. In this way, the spatial relationship is effectively introduced to help the network attain correspondence between the model geometry and the flow field distribution.

Our deep learning method is highly universal, which is not limited to guiding the implementation of CABG and the treatment of CHD. It can analyze and reproduce the relationship between complex cardiovascular geometry and hemodynamics in a given dataset, which can be extended to the hemodynamic simulation of other organs and tissues, or even the flow field research under experimental conditions, such as replacing the steady flow 3D PIV experiment with sufficient data. From a technical perspective, our deep learning method is highly practical. For different properties of hemodynamic parameters, the prediction can still be completed without adjusting the network structure, which was difficult to achieve in the past[28,29]. The analysis results of the velocity and pressure fields confirm that the same network structure can achieve high accuracy predictions for physical fields with and without spatial components. In addition, point cloud, as a conventional data format, can be accepted by most of analysis software (e.g., ANSYS and Python) which makes the processing of point cloud data relatively easy. In terms of resolution, universality, accuracy, and computational efficiency, our deep learning method can meet the needs of most situations. We also noticed that the four datasets (preoperative, postoperative, velocity, and pressure fields) need to be trained separately as inputs, which increased the computational cost and operational complexity of deep learning to a certain extent. Therefore, we will explore potential improvements due to similarities in features between the different fields and application scenarios in further work. For example, by merging four datasets (with different labels), all prediction results can be output in one training session.

The biggest limitation of this study is the lack of clinical data. In CFD simulation, there is no boundary condition information for patients. Currently, we adopted constant values on inlets and outlets, which have been widely used among a number of geometries[44–46]. Therefore, the simulation results should include differences from real hemodynamics. In future approaches that include boundary conditions, another input channel will be required on the network. This input channel will take the patient's personalized boundary conditions as the input, and together with the model point cloud as the teaching signal to control the training process. In the analysis of prediction accuracy, we only compared the prediction results of deep learning with CFD, but lack of comparison with clinically measured data of patients (such as invasive FFR). Itu et al.[47] and Tesche et al.[48] proved that under the premise of good consistency between the FFR calculated by deep learning and CFD, compared with the invasive FFR, the FFR values calculated by these three methods were also with good consistency, which we intend to address in the future. The data for this study comes from a project optimizing the treatment plan of coronary stenosis. Therefore, our datasets do not contain information on other cardiovascular diseases such as coronary aneurysms or aortic diseases. In addition, the point cloud data used in this study is extracted from the CFD meshing result. In the point cloud extraction process, we deleted the connection relationship between the grids. Although the point cloud can reproduce the CFD flow field prediction results, it also brings potential limitations, such as the loss of correlation information between different nodes in the original CFD results and the introduction of the disorder of point clouds. Based on the above limitations and prediction error analysis results, in future work, we need to increase the number and type of patients in datasets to include the characteristics of different cardiovascular diseases and improve the accuracy of predictions. Regarding the datasets with several types of disease, we also need to establish the quantitative methodology to evaluate the variety of geometry as a training data. Secondly, we need to collect physiological information of patients to build the datasets under personalized boundary conditions. Based on this study, we aim to build a network with multiple constraints, multiple channels of input, and multiple sampling layers in parallel. It can help us use

deep learning to achieve the prediction of high-dimensional flow field such as fluid–structure interaction (FSI), etc. Thirdly, the uncertainty of vessel wall identification should be noted as the common limitation in image-based analysis including CFD. Present study exhibits the flow estimation on the point clouds which generated on segmented blood vessel. Then, the flow field strongly depends on the quality of vessel wall segmentation. Though CFD results from the same STL file can exhibit good consistency among different research groups[49], still segmentation process from DICOM images can lead to variability in geometry[50]. To overcome this uncertainty of real geometry, the establishment of stable segmentation method or normalization of hemodynamic parameters will be required.

## Methods

**Ethics approvals.** The experimental scheme and related details of this study were approved by the Institutional Ethics Committee of People's Hospital (Beijing, China) and Tohoku University (Sendai, Miyagi, Japan). All experiments were carried out in accordance with relevant guidelines and regulations. We explained research content to the subjects in detail and obtained their written informed consent.

**Data collection.** The patient data used in this study was based on the project 'Biomechanics study on quantitative relationships between coronary artery stenosis and myocardial ischemia[51–53], which focused on the diagnosis and optimization of coronary stenosis surgical procedures. The CTA data for 110 patients with LAD stenosis who had visited the People's hospital since 2018 was collected and collated by professional clinicians with a 128-slice CT scanner (Brilliance iCT, Philips Healthcare, The Netherlands). 3D model reconstruction was also performed by the clinician. We obtained 110 STL cardiovascular models as raw data.

**Model geometric parameters modification.** The deep learning dataset, which only contained 110 real cardiovascular models, had a very limited amount of information, which was far from enough to represent the relationship between the geometry of the model and the corresponding hemodynamics. Therefore, based on the statistical results of previous cardiovascular morphology studies[47,48,54–59], the geometric parameters of the 110 original cardiovascular models were adjusted to increase the number of models. For each parameter, we randomly selected one value within the given range as the modification basis of the original model, as shown in Table 3. Based on this method, we extended one original model into nine new models, which meant that the total number of models increased to 1100, as shown in Supplementary Fig. 2.

**Simulated operation of CABG and CFD simulation.** After model expansion, we performed the simulation implementation of the CABG operation and the CFD simulation.

As the most patients did not have undergone CABG surgery, the virtual bypass surgery was performed except for undergone CABG case (as seen in Supplementary Fig. 3). With the agreements of clinicians, the left internal mammary artery (LIMA) with diameter of 2 mm was deployed using modeling software Mimics (Materialize NV, BE).

According to the generation of geometry, tetrahedron numerical meshes with boundary layers were generated by ANSYS-Meshing (ANSYS, Canonsburg, USA).

Total mesh number was selected to have the number of nodes from 2.83 to 3.01 million based on mesh-independence test.

Steady flow simulation was performed on ANSYS-CFX (ANSYS, Canonsburg, USA). Blood flow was simplified to be an incompressible Newtonian fluid with 1050 kg/m³ density and 0.0035 Pa·s viscosity. Velocity boundary of 1.125 m/s was imposed on the inlet assuming the peak wave velocity of cardiac cycle[60]. Outlet boundary was set as zero pressure condition. No-slip condition was assigned to all wall boundaries.

More detail is summarized in the Supplementary Method.

**Creation of datasets and proposal of deep learning network.** Using simulation software (e.g., ANSYS, OpenFOAM, etc.), the high-density 3D point cloud form of the four groups—preoperative, postoperative, velocity, and pressure fields—of the CFD simulation results could be directly output (i.e., they could be represented as a set of points $\{P_i \mid i = 1, ..., N\}$ in space). Each point $P_i$ was a vector containing spatial coordinates (x, y, z) and hemodynamic parameters at that point. $P_i$ was the connection point of CFD meshes (usually called node). CFD meshes generation was only related to the geometry of the model. Therefore, the distribution of points depended only on the geometry of the model. The position of points in the model was fixed, we could not change its spatial distribution. What we could do is to directly extract and analyze the points in a certain position through the simulation software.

We divided each group of point cloud data into two sets: a training set and a test set. The training set included simulation results of 1000 cardiovascular models based on the original 1000 models. In order to ensure the generalization of the deep learning network, the test set consisted of the CFD results of 100 cardiovascular models that were expanded from the 10 original models which were different from the training set. Based on this, the four groups—preoperative, postoperative, velocity, and pressure fields—of hemodynamic datasets were established, respectively. These four datasets were used independently to train four separate networks. Hence, we obtain four optimal network configurations to further predict the corresponding hemodynamics.

In the case of certain boundary conditions, the values of flow velocity and pressure at each point were jointly determined by the overall shape of the model and its specific spatial coordinates. This was also the basic principle for CFD to resolve the simulation results via the Navier–Stokes and continuity equations. The segmentation network structure of PointNet[37] could realize feature extraction and hemodynamic prediction of point clouds. This study inherited the concepts of global feature and local feature proposed by the original PointNet, and optimized the network structure. Since the original PointNet had only one single input channel, global features, and local features were extracted from the same and all input points, which could help the PointNet identify the relationship between these two features. However, it was inevitable that there would be duplication between the two features, and then some effective and specific information would be lost. In order to solve this problem, a network structure with double input and double sampling channels was proposed in this study. The structure and parameter setting were shown in Fig. 5. For each model in the dataset, we extracted two types of point clouds. One was the model point cloud, which only included spatial coordinates for the outermost points of the cardiovascular model. These points could represent the global features of the overall model geometry. The second was the query point cloud, which included the remaining points inside the cardiovascular model. These points contained local features, such as the spatial coordinates of each point and its corresponding hemodynamics. The 3D deep learning network had two independent input channels that corresponded to these two point clouds. Two feature extraction parts were directly connected to their respective input and sampling channels. This effectively enhanced the extraction of effective and specific information from these two features, and improved the prediction accuracy. For

**Table 3 Geometric parameters with corresponding ranges.**

|  | Geometric parameter | Parameter details or measurement methods | Range |
|---|---|---|---|
| Coronary artery | Number of branches[47,48] | Main branches | 3 (LAD, LCX, RC) |
|  |  | Side branches | 0–3 |
|  | Bifurcation angle between LAD and LCX[54,55] | The angle described by the two branches in the first 10 mm of their course was measured | 30–90° |
|  | Stenosis location | Random positions on LAD | LAD (main branch) |
|  | Number of stenosis | Determined by the patient's actual condition | 1–2 |
|  | Stenosis Rate | Idealized stenosis model | 60–90% |
| Aorta and superior aortic branch artery | Aortic arch angulation[56,57] | Angulation of the arch at the level of the left subclavian artery | 80–140° |
|  | Diameter of ascending aorta[58,59] | Increased or decreased the diameter of original artery uniformly | 20–30 mm |
|  | Diameter of descending aorta[58,59] | Increased or decreased the diameter of original artery uniformly | 15–20 mm |
|  | Superior aortic branch artery | Kept the original geometry |  |

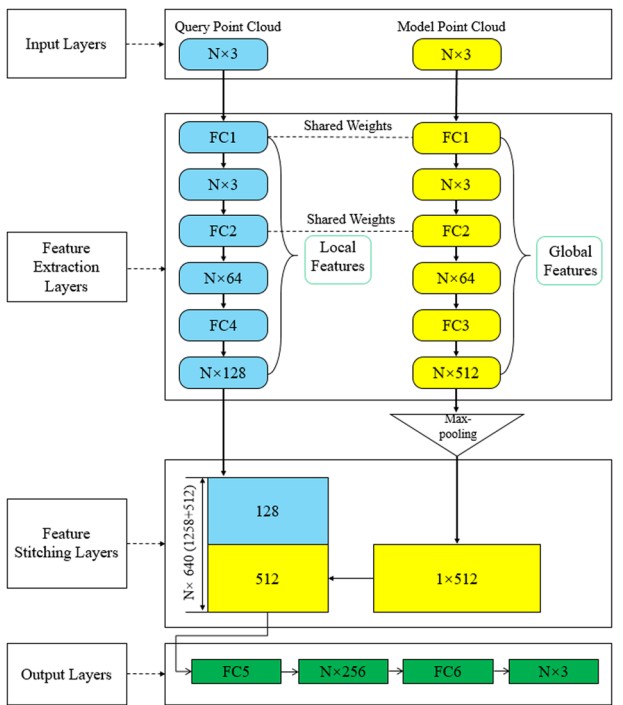

**Fig. 5 Deep learning network construction.** The network takes N points with three-dimensional spatial components of x, y, and z (for pressure, the three-component values are the same) as input. After feature extraction and stitching layer processing, spatial relationships are introduced to extract and reproduce the relationship between model geometry and hemodynamic. The Maxpooling layer resolves the disorder of the point cloud. The output of the network is the hemodynamic three-dimensional spatial components of N points in corresponding query point cloud.

**Table 4 Comparison of MRE from original PointNet and our network.**

| Hemodynamic | Velocity field | Pressure field |
|---|---|---|
| Original PointNet | 18.42 ± 6.71 | 14.59 ± 5.31 |
| Our network | 9.77 ± 3.86 | 7.61 ± 1.99 |

The MRE is calculated according to the hemodynamic prediction values of the preoperative aorta and superior aortic branch artery.

the same point cloud data, the comparison of prediction MRE from the original single channel PointNet and our dual channel network was shown in Table 4. The prediction ERR of our dual channel network was lower than that of the original PointNet.

Global features were the global geometric information of the model. Local features referred to the location of each point and the corresponding flow field distribution inside the model. These two features both contained the geometric features of the same cardiovascular model (commonality). The two features also had different effective information (difference). The network needed to extract commonality and difference and learned the correlation between them to further realize the flow field prediction.

Based on the above principle, the network construction scheme was as follows:

To enhance the commonality and correlation, the first two feedforward fully-connected layers (FC1 and FC2) of the two feature extraction sections shared weights, which meant they shared the same underlying feature extraction methods. In order to evaluate the necessity of sharing weights, we compared the network without shared weights with the results of this study, as shown in Table 5. The results showed that the shared weight could effectively reduce the prediction error. The two feature extraction sections also had independent feedforward fully-connected layers (FC3 and FC4), which further enhanced the ability of the network to recognize the effective and specific information (difference) of global features

**Table 5 Comparison of MRE from network with or without shared weights.**

| Hemodynamic | Velocity field | Pressure field |
|---|---|---|
| Without shared weights | 16.37 ± 5.43 | 13.42 ± 5.21 |
| With shared weights | 9.77 ± 3.86 | 7.61 ± 1.99 |

The MRE is calculated according to the hemodynamic prediction values of preoperative aorta and superior aortic branch artery.

and local features. After FC3 and FC4, the global and local features contained in the two point clouds were represented as a N * 512 and N * 128-dimensional vectors, respectively. We concatenated the two vectors to form an N * 640-dimensional vector. This vector contained both the global features of the model point cloud and the local features of the query point cloud, which helped the network further integrate the correlation between the two features. The last part of the network was feedforward fully-connected layers (FC5 and FC6), which were used to yield hemodynamic results.

For other details of the network, we added a Maxpooling layer as a symmetric function in the feature extraction part of the model point cloud, which could aid in resolving the disorder of the input point cloud[37]. We used the mean absolute error as the regression loss function[24,61]. We used the Adam optimizer with specific parameters: learning rate = 0.001, $\epsilon$ = 0.001, $\rho1$ = 0.9, $\rho2$ = 0.999, and $\delta$ = 1E −8[62]. The 3D deep learning network was trained by TensorFlow (v2.0.0rc, Python 3.6 on a Nvidia GeForce GTX 1080 Ti GPU). The preoperative and postoperative datasets needed to be separately trained as inputs for the network. During the training process, we saved the optimal network parameter configurations for both training sets. After that, while only inputting the spatial coordinates in the test set, the network could recognize the hemodynamic prediction output of query point cloud.

**Definition of error functions.** Referring to the evaluation criteria of previous studies, NMAE[29] and MRE[28] were defined as error functions to evaluate the accuracy of deep learning network predictions based on the CFD results, as shown in equation (1) and equation (2):

$$\text{NMAE} = \frac{1}{N} \frac{\sum_{i=1}^{N} |P_i - \hat{P}_i|}{\text{Max}|P| - \text{Min}|P|} \times 100\% \qquad (1)$$

$$\text{MRE} = \frac{1}{N} \sum_{i=1}^{N} \frac{\sqrt{\left(P_i - \hat{P}_i\right)^2}}{\sqrt{P_i^2}} \times 100\% \qquad (2)$$

$N$ represented the number of selected query points. $i$ was the spatial sequence of the 3D point cloud. $P_i$ and $\hat{P}_i$ represented the flow velocity or pressure value at a certain point calculated by CFD and deep learning, respectively. $\text{Max}|P|$ and $\text{Min}|P|$ represented the maximum and minimum magnitude of the corresponding hemodynamic parameters among all points in the selected area, respectively. NMAE can characterize the error of the deep learning prediction result relative to the true value of the overall flow field (CFD result). MRE can characterize the error of the deep learning prediction value relative to the true value at all query points of the model. The definition of the error function draws on previous studies. The comparative analysis results are shown in Supplementary Table 1. In this study, ERR is designed to evaluate the velocity or pressure fields represented by point clouds. For other parameters (such as FFR calculated by pressure field, etc.), new ERR should be defined according to the specific situation. In these definitions, each of the points of different cardiovascular parts can affect ERR with the same weight. However, the number of points and the magnitude of velocity and pressure must have a great difference among aorta, coronary arteries, and bypass graft. In order to avoid the impact of this variation on the evaluation results, local ERR (The model was divided into several parts, and the ERR value of a certain part, such as the left anterior descending branch, was called local ERR.) values were obtained to assess the prediction accuracy on small-to-large parts. We calculated the ERR values of the proximal and distal end of left anterior descending artery (LAD), graft, right coronary artery (RA), the left circumflex branch (LCX), the aorta and superior aortic branch artery, respectively. Regarding the LAD, the proximal and distal ends were divided by stenosis. When there were multiple stenosis in the LAD branch, the stenosis with highest degree was selected.

**Statistics and reproducibility.** All ERR calculations were based on the velocity or pressure results of 100 models in the test set. The definition of query point cloud number ($i$) was defined in equation (1) and equation (2). This study took the average value of ERR of 100 models. The standard deviation was used to calculate the error bars.

**Reporting summary**. Further information on research design is available in the Nature Research Reporting Summary linked to this article.

## Data availability

Data analyzed during the current study are available from the corresponding author upon reasonable request. Restrictions apply to the sharing of patient data that supports the findings of this study. With the approval of the Institutional Ethics Committee of People's Hospital, the patient's data can be authorized for use by qualified researchers. The Source data underlying the graphs and charts presented in the main figures (from Fig. 1 to Fig. 4) can be accessed at: https://doi.org/10.6084/m9.figshare.13295915.v1[63].

## Code availability

All source code described in this project can be accessed at: https://doi.org/10.5281/zenodo.4287103[64].

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

## Acknowledgements

This research is partially supported by the Creation of a development platform for implantable/wearable medical devices by a novel physiological data integration system of the Program on Open Innovation Platform with Enterprises, Research Institute and Academia (OPERA) from the Japan Science and Technology Agency (JST). This work is also supported by the JSPS KAKENHI with the Grant Number JP18K18355, the Grant-in-Aid [A] (No16H01805), the Grant-in-Aid [C] (17K01444), the Grant-in-Aid [C] (19K04163), the National Natural Science Foundation of China (11772015), and the National Natural Science Foundation of China (11832003, 11772016).

## Author contributions

G.L., H.W., Y.L., and A.Q. acquired the data. G.L., H.W., M.Z., S.T., A.Q., M.O., Y.L., and H.A. created and designed this study; G.L., H.W., M.Z., S.T., and H.A. performed the experiments and analyzed the data. All of the authors discussed and co-authored the manuscript. The contributions of G.L. and H.W. were equal.

## Competing interests

The authors declare no competing interests.
