## [Peer Review File · Communications Biology]

Reviewers' comments:

Reviewer #1 (Remarks to the Author):

The authors of this article describe a proposed deep learning approach to estimate the 3D hemodynamics of complex aorta-coronary artery geometry in the context of coronary artery bypass surgery. They address the computational cost of traditional 3D CFD methods by developing a deep learning (DL) workflow, which they claim can generate a patient's prediction result in 1 second after training, a 600-fold increase in speed. The methodology appears to be robust to predefined variations in geometry and consistent with traditional CFD. The main innovation leading to outperformance over existing similar studies is the point cloud structure allowing for maintenance of high resolution and accuracy of predicted hemodynamics within complex vascular geometry.

While the computational performance of this approach is impressive, it is unclear to what extent the DL method is expected to transform diagnosis and treatment of coronary artery disease over gold standard FFR measurement. How much of global coronary heart disease morbidity and mortality is attributed to insufficient knowledge of 3D hemodynamics? Furthermore, the extent of validation of DL prediction accuracy in this paper is comparison with traditional CFD, but even traditional CFD has its assumptions that are not physiological. This is concerning because Nature Communications Biology "represent significant advances bringing new biological insight," but my impression overall is that this is a very methodology-heavy paper that may be more appropriate for Nature Methods than Nature Communications Biology unless more biological/clinical validation is added. Some sort of catheter flow/pressure measurement of the patients undergoing treatment would significantly strengthen the evidence supporting the authors' claims.

See below for a comprehensive list of technical comments.

Major Comments:

1. The authors' assessment that "obtaining hemodynamics for operational support of CABG" is important (lines 35-37) should be made stronger by describing the current clinical unmet need that can be addressed with 3D hemodynamic prediction.
2. Accuracy of 3D hemodynamics prediction is defined as agreement with CFD. In order to support the argument of clinical reliability of the DL approach, the gold standard should be catheter measurement of FFR, pressure, and flow in patients.
3. The authors also claim that the DL method will help diagnose severity of coronary ischemia (lines 103-104). Validation of this using clinical data corresponding to patient's geometry such as cardiologist assessment has not been presented in this study.
4. Reference CFD method needs to be described in detail in Methods section.

Technical Comments, Clarifications and Suggestions:

1. It appears that an important step in addressing the methodological challenge of DL on patient-specific complex 3D hemodynamics is that "new requirements have been imposed" (line 71). It is unclear whether these requirements have been imposed by multiple research groups in the field (in which case, please cite references) or by the authors (in which case, please clearly state that this is the contribution of the paper)

2. Lines 103-104: "Our deep learning method will link to effectively diagnose..." –Should "link" be changed to "aim"?
3. Additional recent articles that may be relevant for citation: Coenen et al. 2018 (PMID: 29914866) and Wang et al. 2019 (PMID: 30800150)
4. It is not obvious how your model would incorporate patient specific boundary conditions. This comes from the fact that you create your dataset using a CFD solver, in which you have to impose boundary conditions. This is not an easy feat, so if you want to claim that your method is generalizable please elaborate more. How will the additional input channel be used?
5. You claim that layers FC3 and FC4 helped the network identify the differences between global features and local features. Why is that? Could you elaborate more? What information do you have that point to this direction?
6. You claim that your method is robust, but you do not present any study that points into this direction. Robustness in neural networks means that your network using different seed to initialize the weights provides predictions that are close to each other. Please support your claim by providing a systematic study.
7. For figure 2 please make the units to be in \$mmHg\$. Also, the pressure scale seems off because the pressure in the aorta is usually 60-110 \$mmHg\$, but what you show is around 6 \$mmHg\$. Please provide correct scales, because these values are not physiological, so you cannot claim that your method has clinical applicability if your predictions are 2 orders of magnitude off.
8. For figure 1j C: Why are the cross-sections different between CFD and DL? You are using only the query points to predict the velocity, so how are the wall points affected?
9. For figure 1 in supplementary material: Please make the plot in logarithmic scale to easier to read.
10. For supplementary discussion: It seems that the method proposed by Liang et al. is performing in a similar fashion as yours. Why is your method better? Please provide more information and be specific on the problem setup of the other methods and the advantages of your method. Explain thoroughly.
11. It is confusing to me how come that when you use 100% of your dataset in the vortex region you get 28% error and when you are using 30% you are getting less. I believe that what you are trying to show is clear, but the method is confusing. Please explain more thoroughly and clearly your procedure.
12. It is also not clear how you make predictions. I understand that you train the same network on post and pre surgery data, it is not clear to me if then you provide a post and pre surgery geometry characteristics to the same network and make predictions. Please explain.
13. I understand that you enrich the data set by making random changes to the geometry characteristics, i.e. stenosis rate, but how does this affect the arterial geometry(i.e. the shape of the arterial walls)? Do you provide a different geometry or do you provide the same geometry, but with different characteristics? Please explain.
14. Could you provide some insight on how close is the test data geometry to the training set? Is it significantly different or close? This will help you show that the network can generalize to different architectures.
15. line 43: In my opinion, you should mention at this point the use of reduced order models in cardiovascular hemodynamics and especially in the case of predicting FFR. The reduced order models are inexpensive and accurate in some situations(see *Estimating the accuracy of a reduced-order model for the calculation of fractional flow reserve (FFR).*). Why is it necessary to employ a 3D model of Navier-Stokes? What are the geometry characteristics that prohibit the use of reduced order models?
16. line 66-67: In my understanding you present these common CFD limitations to strengthen your argument about employing deep learning techniques for predicting the flow characteristics, but

there are some points that you make that are not clear to me. For 2), if I understand correctly you are implementing your model to a steady state incompressible Navier-Stokes equation with constant boundary conditions, where the flow data are acquired by a conventional CFD simulator and a virtual surgery. This makes the flow pretty ideal, because the difference between ideal and patient measured flow is the presence of noise which makes the measurements not necessarily a solution to your system of equations. So, my first question is: What makes your set-up non-ideal and difficult to the point of proposing a new architecture and implement deep learning? Do you use both patient specific and data acquired by ANSYS or only data from ANSYS? Please elaborate.

17. line 66-67: For 3): You claim that one point that makes your method superior in comparison with other methods is the use of a small dataset and other models need millions of samples. Please elaborate more on what you mean by small and large datasets. In your case, if I understand correctly you use 1100 geometries each consisting of 2 million points, which corresponds to million of samples. Do you mean that other studies require million of geometries?

18. line 66-67: Another point you should consider is the wall identification noise, which is a significant parameter of uncertainty in hemodynamics predictions in clinical practice, and how would this affect your method. Please provide some insight on that, because your method is relying on wall points from a mesh, which makes this a valid point.

19. line 88-90: It is not obvious what you mean by "that even high-density point clouds can store a great deal of valid information with little data". Do you mean that you do not need to store the connectivity information? What make this method of storing to incorporate information in a sense that make this superior over other/ Is this just the memory capacity?

20. line 113:: This is not correct. You cannot assess overfitting from the loss function value over epochs for the training set. Overfitting means that the model can make accurate predictions for the training set and not for the test set. If you provided a figure with the error on the test set during training then you probably mention overfitting.

21. line 127: I believe that this is a misleading point, because you do not explain why you model cannot make accurate predictions in the region where the vortex is occurring and also you do not propose any improvement, you just show that if you assign more points in the region you can get better accuracy, which is not considered and improvement to you model. Also, 28% error for 2 million samples is not considered very accurate.

22. line 141: I believe that you are referring to pressure being a scalar field. Please be a little bit more descriptive, because this might not be clear to the reader.

23. line 144: See above.

24. It is not clear to me how can the two pressure distributions be close to each other granted that for incompressible Navier-Stokes you can only predict the absolute pressure up to a constant. Can you provide some insight regarding that?

25. It is not clear to me if the Bland-Altman analysis proves your point of the data not having any systemic bias. In figure 3d, you can see that there is in fact some bias, if I understand correctly you plot, which is not very small considering the flow order of magnitude(around 10%).

26. line 208: Could you provide more information regarding these cases of vortex prediction that you are referring to? What was the setup? How was it similar or different than yours?

27. line 238: Please explain where do these numbers come from. Why $800 \times 1200 \times 950$?

28. line 252-254: Please explain what you mean by spatial disorder. The readers might come from a diverse background, so this might not be obvious to them.

29. line 258: "local feature... teacher's signal" I not sure what you mean at this point. Please rephrase or elaborate more.

30. line 386-387: What do you mean by local ERR values? Could you elaborate more?

31. line 391: Can there be multiple stenosis in your LAD branch? Is this a part of the data set? Please be more specific about your dataset creation because it might be confusing to the reader.

Grammar/Typos:

1. line 36: ...obtaining hemodynamics. Please rephrase because it is not clear what you mean. Hemodynamics is a very general term.
2. line 41: ...hemodynamic numerical models. Please rephrase this is not correct grammar.
3. line 55: ...prediction. I believe that a more accurate term would be regression.
4. line 56: ... the computing power of deep learning. It is not clear to what you are referring to and what you mean by the term computing power.
5. line 57: ...expansion. I believe that this is not a proper term to describe what you mean. Please correct the language.
6. line 76: "Based on the above background": Please rephrase.
7. line 78: "at once": Please rephrase.
8. line 79: "velocity and pressure field\textbf{s}"
9. line 81-82: "simulation implementation". Please rephrase.
10. line 82: "the CFD method". Change to "a CFD method"
11. line 89: "even high-density...valid information". Probably you need to remove even and valid.
12. line 94: "optimal weight assignment": Please rephrase.
13. line 98: "we defined.." Please change this to "we define"
14. line 112: "learning curve": You should probably rephrase that to "Loss function value versus epochs"
15. line 116-117: "It was confirmed ..." Please rephrase.
16. line 124: "mainly due to the part". Please rephrase.
17. line 162: "in the test set was input". Please rephrase.
18. line 172: "had broad application..". Please change to "has ..."
19. line 195-196: Please rephrase. The grammar is not proper.
20. line 214: "dataset is very necessary". Remove "very".

Reviewer #2 (Remarks to the Author):

Summary:

The paper addresses the challenge of non-invasively predicting hemodynamic parameters from imaging data. Clinically approved methods for this purpose involve computational fluid dynamics, which bear limitations such as extensive computation times and sensitivity to boundary conditions. This paper follows the recently emerging approach of approximating simulations with deep learning models. For this purpose, authors propose a novel network architecture which builds upon the highly recognized PointNet architecture. PointNet is capable of directly processing pointclouds, which correspond to the input and output format of simulations.

The sparse nature of pointclouds, enables direct processing of the simulation input data corresponding to the whole system of aorta and coronaries without compression. The model is trained and evaluated on pointclouds extracted from in-vivo image data.

Evaluation shows good agreement of model output with the simulated prediction targets, in-line with performance of previous work.

General Comment:

The paper is well written and structured with a few spelling and grammar mistakes (see minor issues). The presented results do not show any methodological flaws.

However, clarity of the motivation and discussion of the deep learning architecture needs improvement. At the current stage it is cumbersome to assess which parts of the proposed architecture are novel.

While results and motivation support the claim of superiority of PointNet-based architectures, unclarities remain in the motivation of the architecture.

Another main weakness lies in missing evaluation on in-vivo measured hemodynamic parameters as only simulated parameters are used for training and testing throughout this work.

Originality and significance:

Application of a PointNet-based algorithm to hemodynamic simulations has not been done before to the best of my knowledge. This application is interesting because it enables direct application of the model as a surrogate for computational fluid dynamics.

The sparsity of pointclouds enables the given approach to process the entirety of information associated with the combined system of aorta and coronary arteries.

Thorough evaluation regarding the origin of most erroneous predictions supports the claim of general applicability of the method to other prediction scenarios involving simulations.

However, shortcomings in the evaluation of the presented network architecture prevent conclusions about utility of the presented architectural novelties with respect to the original PointNet.

Major Issues:

1

Changes made to the original PointNet architecture were not explicitly stated.

While the methods section (in the end of the paper) enables identifying these changes, they should be stated explicitly.

The original PointNet architecture also incorporates local and global features, however both derived from the query points. Please motivate the separate model branch considering this.

More severely, these changes were not explicitly evaluated. To strengthen justification of the model branch, I propose an ablation experiment where the original PointNet is directly applied.

Furthermore, shared weights impose a limitation to the expressivity of the network. It would be interesting to evaluate beneficiality of shared weights in the first two layers by comparing results against a training run without shared weights.

Current absence of these evaluations makes some of the claims in the discussion seem unsupported, i.e. lines 255 - 260.

2

While utility of global (point 1) and local (point 2) features is intuitive, point 3 in the desired functions of the algorithm needs further clarification.

336: "The ability to learn and reproduce the relationship between two features, especially regarding commonality, difference, and correlation."

Does "two features" refer to local vs global features? Please briefly explain why commonality, difference, and correlation is a required learning goal.

3

"Concerning the dataset, each sample must have enough spatial resolution to resolve complex flow

field and model geometry.

And concerning the network, enhancing its feature extraction capabilities would be necessary so that it could

efficiently obtain valid information with a limited sample number."

Please specify what exactly you mean by enhanced feature extraction capability. This should be picked up in the methods section. How do you achieve what you mean by enhanced feature extraction capabilities?

4

The following two statements seem to contradict each other.

1. (line 43) "When CFD is used to calculate the hemodynamics of a complex cardiovascular model with small grafts and coronary branches, even high-performance computing clusters will usually need several hours of iteration to ensure model accuracy."

2. (line 163) "For the CFD method, the calculation time of one model on an Intel Xeon Gold 6148 2.4Ghz × 2 CPU server was about 10 minutes."

Please elaborate.

5

"Compared to other 3D data formats (e.g., voxel grids), the point cloud format has a simple and unified structure. It does not introduce irregular shape and connection information, which means that even high-density point clouds can store a great deal of valid information with little data"

Please explicitly establish the link between data irregularity and compressibility.

6

Limitations of this work with respect to previous work seems to be missing a key aspect. While a model operating on point-clouds is ideal for replicating simulation results, in-vivo data is structured on a grid and discarding this structure in favor of a point cloud erases potentially relevant relational information. Please elaborate on this potential limitation.

7

21 (Abstract): "Our deep learning method is significantly better than existing deep learning approaches..."

Please be more specific on what is better, e.g. accuracy, computation time, applicable regions. The comparison to previous work in the supplement does not support this claim, as similarly well performing methods are listed (i.e. Liang 2020).

8

112: "...the learning curve was made available (as seen in Supplementary Fig. 1). The loss function fully converged without overfitting."

Does this learning curve correspond to evaluation on the training or on a separate validation set? Convergence on training data does not enable outruling overfitting, which typically only shows up on the validation set.

Minor Issues

364: We used the mean absolute error as the regression loss function, which made the network more robust to outliers in the input data.
Please explicitly state the alternative.

368: "The preoperative and postoperative datasets needed to be separately trained as inputs for the network. During the training process, we saved the optimal network parameter configurations for both training sets."
This explanation seems to imply that two separate models were trained. Please state if this was the case or not.

59: "For example, Guo et al. put forward a calculation method of flow around simple geometric models based on convolutional neural networks. And Liang et al. proposed a deep learning method to predict simplified thoracic aortic hemodynamics."
It is not clear why these two examples were chosen from amongst the references that are subsequently discussed regarding their limitations.

Table 1.

The headline "The error functions of the velocity field" gives the impression that the table describes the analytic function itself, while it actually presents evaluations of this function. I would propose replacing "functions" with "metrics" or "Performance evaluation of the velocity field".

377: Please discuss suitability of both loss functions for this specific task

391: "...the highest stenosis rate was selected."  "... the stenosis with highest degree was selected."

148: "...pressure distribution in of the..."  "...pressure distribution of the..."

166: "..., this process only need to be completed..."  "..., this process only needs to be completed..."

206: "...much more training set data than ours, ..."  "...much more training data than ours, ..."

244: "...generally called as node."  "...generally called a node."

RE: Manuscript ID COMMSBIO-20-1086-T

Dear editor and dear reviewers,

We thank the editor for giving us the opportunity to submit a revised manuscript and the reviewers for their evaluations. We are grateful to all reviewers for their time and highly constructive comments. We try our best to address the concerns you raised in the form of our revised manuscript. All the changes we made are highlighted in the revised manuscript. This document lists the new parts point by point in the way of question and answer. We hope we could answer everything to your satisfaction and are looking forward to your feedback.

Best wishes,

Hitomi Anzai

Corresponding Author

Reviewer#1

Remarks to the Author: the authors of this article describe a proposed deep learning approach to estimate the 3D hemodynamics of complex aorta-coronary artery geometry in the context of coronary artery bypass surgery. They address the computational cost of traditional 3D CFD methods by developing a deep learning (DL) workflow, which they claim can generate a patient's prediction result in 1 second after training, a 600-fold increase in speed. The methodology appears to be robust to predefined variations in geometry and consistent with traditional CFD. The main innovation leading to outperformance over existing similar studies is the point cloud structure allowing for maintenance of high resolution and accuracy of predicted hemodynamics within complex vascular geometry.

While the computational performance of this approach is impressive, it is unclear to what extent the DL method is expected to transform diagnosis and treatment of coronary artery disease over gold standard FFR measurement. How much of global coronary heart disease morbidity and mortality is attributed to insufficient knowledge of 3D hemodynamics? Furthermore, the extent of validation of DL prediction accuracy in this paper is comparison with traditional CFD, but even traditional CFD has its assumptions that are not physiological. This is concerning because Nature Communications Biology “represent significant advances bringing new biological insight,” but my impression overall is that this is a very methodology-heavy paper that may be more appropriate for

Nature Methods than Nature Communications Biology unless more biological/clinical validation is added. Some sort of catheter flow/pressure measurement of the patients undergoing treatment would significantly strengthen the evidence supporting the authors' claims.

Thank you very much for your comments. Based on your comments, we have made the following amendments to the manuscript and explained in detail in the answers to specific questions below:

“While the computational performance of this approach is impressive, it is unclear to what extent the DL method is expected to transform diagnosis and treatment of coronary artery disease over gold standard FFR measurement.”

Answer:

Thank you very much for your comments. We totally agree with your comment about the gold standard FFR. However, the main purpose of this study is to show our deep learning method has a similar function to computational FFR method (FFRct). At the same time, our deep learning method can overcome the high computational cost of computational FFR method.

However, based on your suggestions, we realized the necessity of descriptions about the gold standard FFR and the followings are added:

1. We add a description of the clinical shortcomings (**Introduction Line36: “Taking invasive FFR as an example, the price of the pressure guide wire required for measurement is relatively high. In addition...”**) of the gold standard FFR to prove that it is necessary to develop a deep learning method for the calculation of FFR.
2. FFR can help us further analyze the difference between the pressure field calculated by deep learning and CFD method. Based on the limitations of the current research, we can only say that the FFR values calculated by deep learning and CFD have good consistency. The validation using catheter measurement will be included in the next step of our research. We further explained it in the **limitations section of the Discussion**.

“Furthermore, the extent of validation of DL prediction accuracy in this paper is comparison with traditional CFD, but even traditional CFD has its assumptions that are not physiological.....Some sort of catheter flow/pressure measurement of the patients undergoing treatment would significantly strengthen the evidence supporting the authors' claims.”

Answer:

Thank you very much for your comments. Based on your comments, we made the following changes in the manuscript:

1. Clarify the main research purposes in **Introduction**:

- 1) This study aims to develop a deep learning method to realize the fast prediction of velocity and pressure field of cardiovascular system to reduce the high computational cost of CFD.
- 2) This study aims to develop a deep learning approach that can realize 3D personalized cardiovascular system hemodynamic prediction, in view of the fact that previous deep learning methods are only suitable for 2D or ideal models.

2. Clarify the limitations of the current study in **Discussion**:

Limited by the patient's clinical data, this study lacked information on the flow/pressure measured by the catheter. We only compared the results of CFD and deep learning. We give a detailed statement on this point in the **limitations section of the Discussion**. We fully agree with you that “but even traditional CFD has its assumptions that are not physiological”. **Despite the shortcomings of the CFD method, a large number of studies have shown that the CFD method can still play a guiding role in the treatment of CABG^{1,2,3,4}**. For example, FFR_{ct} calculated by CFD method, this non-invasive detection index has been confirmed to have good consistency with invasive FFR and is widely used in clinical^{5,6,7}. **On this basis, we believe that our research still has some clinical significance**. In addition, Itu et al.⁸ and Tesche et al.⁹ proved that under the premise of good consistency between the FFR calculated by deep learning and CFD, compared with the invasive FFR, the FFR values calculated by the three methods were also with good consistency, which is also one of the tasks we will do next.

“How much of global coronary heart disease morbidity and mortality is attributed to insufficient knowledge of 3D hemodynamics?”

Answer:

To the best of my knowledge, there is still lack of research about the statistical analysis concerning the attribution of hemodynamics to the morbidity and mortality. However, there are a lot of papers talking the relations between hemodynamics and CABG including case reports^{1,2,3,4}. Currently, the formulation of the CABG plan mainly relies on the patient's clinical images, and there are often lacks of hemodynamic guidance. This is because the invasive hemodynamic parameter measurement method is difficult to be accepted by patients due to its high cost and potential risks. On the other hand, the high computational cost of CFD method is not conducive to clinical use. In this study, our deep learning methods make up for the above shortcomings due to its advantages of fast and non-invasive.

Major Comments:

1. **The authors’ assessment that “obtaining hemodynamics for operational support of CABG” is important (lines 35-37) should be made stronger by describing the current clinical unmet need that can be addressed with 3D**

hemodynamic prediction.

Answer:

We are very thankful for your comment. In the revised manuscript, we first describe the limitations of current clinical hemodynamic measurements: “**(Line34)** However, the application rate of hemodynamic parameters in clinical practice is low, mainly due to its high measurement cost and potential risks during catheter insertion. Taking invasive FFR as an example, the price of the pressure guide wire required for measurement is relatively high. In addition, the use of vasodilator drugs such as adenosine may increase the time and cost of interventional surgery¹³, and may also increase the surgical risk of patients with adenosine sensitivity or asthma¹⁴. Therefore, how to obtain hemodynamic parameters including velocity and pressure inexpensively and non-invasively is crucial for the support of CABG and the treatment of CHD.” Then we point out that although “**(Line44)** CFD can inexpensively and non-invasively obtain solutions for velocity and pressure,” “**(Line55)** The high computational cost of CFD hinders its clinical application to the treatment of CHD.” Therefore, using deep learning to achieve inexpensive, non-invasive, and fast “3D hemodynamic prediction” is important “for operational support of CABG”. Please check it.

- 2. Accuracy of 3D hemodynamics prediction is defined as agreement with CFD. In order to support the argument of clinical reliability of the DL approach, the gold standard should be catheter measurement of FFR, pressure, and flow in patients.**

Answer:

We are very thankful for your comment and we totally agree with that. However, due to the limitation of current clinical data, we cannot obtain the corresponding patient invasive FFR data. We have added a description of the limitations of this part in the Discussion section: “**(Line305)** In the analysis of prediction accuracy, we only compared the prediction results of deep learning with CFD, but lack of comparison with clinical measured data of patients (such as invasive FFR). Itu et al.⁴⁷ and Tesche et al.⁴⁸ proved that under the premise of good consistency between the FFR calculated by deep learning and CFD, compared with the invasive FFR, the FFR values calculated by these three methods were also with good consistency, which is also one of the further work we need to accomplish.” Please check it.

- 3. The authors also claim that the DL method will help diagnose severity of coronary ischemia (lines 103-104). Validation of this using clinical data corresponding to patient’s geometry such as cardiologist assessment has not been presented in this study.**

Answer:

We are very thankful for your comment and we apologize for our inappropriate description. According to your comments, we have made clear the purpose and significance of this research in the manuscript, that is, “**(Line107) Our deep learning method aims to realize the prediction of velocity and pressure fields before and after CABG surgery instead of CFD.**” Please check it.

4. Reference CFD method needs to be described in detail in Methods section.

Answer:

We are very thankful for your comment. In addition to the Supplementary Information, we added a description of the CFD method in the main text of the manuscript:

“(Line 358) Simulated operation of CABG and CFD simulation

After model expansion, we performed the simulation implementation of the CABG operation and the CFD simulation.

As the most patients did not have undergone CABG surgery, the virtual bypass surgery was performed except for undergone CABG case. With the agreements of clinicians, the left internal mammary artery (LIMA) with diameter of 2 mm was deployed using modeling software Mimics (Materialize NV, BE).

According to the generation of geometry, tetrahedron numerical meshes with boundary layers were generated by ANSYS-Meshing (ANSYS, Canonsburg, USA). Total mesh number was selected to have the number of nodes from 2.83 to 3.01 million based on mesh-independence test.

Steady flow simulation was performed on ANSYS-CFX (ANSYS, Canonsburg, USA). Blood flow was simplified to be an incompressible Newtonian fluid with 1050 kg/m³ density and 0.0035 Pa s viscosity. Velocity boundary of 1.125 m/s was imposed on the inlet assuming the peak wave velocity of cardiac cycle⁶⁰. Outlet boundary was set as zero pressure condition. No-slip condition was assigned on all wall boundary.

More detail is summarized in the Supplementary Method.”

Technical Comments, Clarifications and Suggestions:

- 1. It appears that an important step in addressing the methodological challenge of DL on patient-specific complex 3D hemodynamics is that “new requirements have been imposed” (line 71). It is unclear whether these requirements have been imposed by multiple research groups in the field (in which case, please cite references) or by the authors (in which case, please clearly state that this is the contribution of the paper).**

Answer:

This “new requirements” comes up by the current research limitation based on the overview of currently research situation. Currently, the study of

hemodynamic prediction via deep learning is relatively limited as stated in **Line69**: “The main limitations of these studies are: 1) most studies focus on 2D flow fields, which have a limited scope of application³¹⁻³⁴, 2) the 3D flow field model only appears in ideal geometry, and sample resolution in the dataset is too low to represent complex flow field distributions and geometric structures^{28,29}.” In this study, 3D hemodynamic prediction of real patient model based on deep learning was completed for the first time. To overcome those “inflexibility and low-resolution of the input geometry” and to apply deep-learning based flow-prediction on complex arterial geometry, “new requirements” is proposed based on the datasets and network of this study, which belongs to the contribution of this study.

As suggested, we stated the contribution of this study: “**(Line74)** Therefore, in this study, in order to accurately predict complex 3D cardiovascular hemodynamics with limited samples, new requirements to adapt a flexibility and high resolution on the input geometry have been imposed on datasets and deep learning networks, which is also the main technical problem and contribution of this study.”

2. **Lines 103-104: “Our deep learning method will link to effectively diagnose...” –Should “link” be changed to “aim”?**

Answer:

We are very thankful for your comment. We revised the word from “will link” to “aims” **(Line107)**.

3. **Additional recent articles that may be relevant for citation: Coenen et al. 2018 (PMID: 29914866) and Wang et al. 2019 (PMID: 30800150).**

Answer:

We are very thankful for your comment. As suggested, we cited two articles you mentioned **(Line62: thus completing the task of classification or regression^{21-25,26,27})**

4. **It is not obvious how your model would incorporate patient specific boundary conditions. This comes from the fact that you create your dataset using a CFD solver, in which you have to impose boundary conditions. This is not an easy feat, so if you want to claim that your method is generalizable please elaborate more. How will the additional input channel be used?**

Answer:

We are very thankful for your comment. Yes, in the original manuscript, we did not add details about the CFD boundary conditions in the main text. We are very sorry for the confusion.

We have added CFD method paragraphs to the revised manuscript to make clear that we imposed constant flow rate among subjects, and we also add some sentence on discussion section about the further progress for patient specific boundary (Line359) “After model expansion, we performed the simulation implementation of the CABG operation and the CFD simulation.

As the most patients did not have undergone CABG surgery, the virtual bypass surgery was performed except for undergone CABG case. With the agreements of clinicians, the left internal mammary artery (LIMA) with diameter of 2 mm was deployed using modeling software Mimics (Materialize NV, BE).

According to the generation of geometry, tetrahedron numerical meshes with boundary layers were generated by ANSYS-Meshing (ANSYS, Canonsburg, USA). Total mesh number was selected to have the number of nodes from 2.83 to 3.01 million based on mesh-independence test.

Steady flow simulation was performed on ANSYS-CFX (ANSYS, Canonsburg, USA). Blood flow was simplified to be an incompressible Newtonian fluid with 1050 kg/m³ density and 0.0035 Pa s viscosity. Velocity boundary of 1.125 m/s was imposed on the inlet assuming the peak wave velocity of cardiac cycle⁶⁰. Outlet boundary was set as zero pressure condition. No-slip condition was assigned on all wall boundary.

More detail is summarized in the Supplementary Method.”

In this study, all the models adopt uniform boundary conditions, rather than personalized boundary conditions of patients.

What we mean by “generalizable” is that our deep learning approach can achieve accurate hemodynamic prediction for cardiovascular models with different geometric structures.

In the discussion section, we further clarify the relationship between “additional input channel” and patients' personalized boundary conditions: (Line302) “In future approaches that include boundary conditions, another input channel will be required on the network. This input channel will take the patient's personalized boundary conditions as the input, and together with the model point cloud as the teaching signal to control the training process.”

- 5. You claim that layers FC3 and FC4 helped the network identify the differences between global features and local features. Why is that? Could you elaborate more? What information do you have that point to this direction?**

Answer:

We are very thankful for your comment. The basis here comes from the structural design of the original PointNet¹¹, which is clearly explained in the revised manuscript: (Line395) “The segmentation network structure of PointNet³⁷ could realize feature extraction and hemodynamic prediction of point cloud. This study inherited the concepts of global feature and local feature proposed by the original PointNet, and optimized the network structure.” (Line427) “The two feature

extraction sections also had independent feedforward fully-connected layers (FC3 and FC4), which further enhanced the ability of the network to recognize the effective and specific information (difference) of global features and local features. After FC3 and FC4, the global and local features contained in the two point clouds were represented as a $N * 512$ and $N * 128$ -dimensional vectors, respectively.”

In addition, to avoid similar confusion, we rewrote the design part of the network. We clearly compared the similarities and optimization schemes between our network and the original PointNet. The reasons for the optimization design were also given in detail (add an explanation experiment and a control experiment). Please check it (Line392-436).

6. **You claim that your method is robust, but you do not present any study that points into this direction. Robustness in neural networks means that your network using different seed to initialize the weights provides predictions that are close to each other. Please support your claim by providing a systematic study.**

Answer:

We appreciate your comment. We are very sorry for the confusion caused by our misunderstanding of “robustness”.

We have revised the manuscript as follows:

1. (Line439) remove the wrong description “which made the network more robust to outliers in the input data.”
2. (Line137) here, we want to show that our network can accurately predict the velocity or pressure field of the corresponding model regardless of whether the "graft" exists or not. We sum it up as “high performance”, that is, “The proposed network could effectively identify significant and non-significant disturbances of the graft on the flow field, which highlighted its high performance.”

7. **For figure 2 please make the units to be in \$mmHg\$. Also, the pressure scale seems off because the pressure in the aorta is usually 60-110 \$mmHg\$, but what you show is around 6 \$mmHg\$. Please provide correct scales, because these values are not physiological, so you cannot claim that your method has clinical applicability if your predictions are 2 orders of magnitude off.**

Answer:

We are very thankful for your comment. As suggested:

1. (Fig.2 Line663) We converted the unit to “mmHg” (as shown following).

2. The “6mmHg” is not the real pressure in the aorta. The “6mmHg” is the pressure difference relative to the coronary outlet. In this study, all the models adopt uniform boundary conditions, rather than personalized boundary conditions of patients **“(Line371: Outlet boundary was set as zero pressure condition.)”** This kind of boundary condition (outlet zero pressure) has been widely used by a large number of CFD studies^{12,13,14}. According to the setting of this boundary condition, what CFD calculates is not the absolute pressure value of the aorta, but the pressure difference (6 mmHg) of the aorta relative to the coronary artery outlet. We give a further explanation in **the legend of Fig.2(Line665: Because the CFD outlet boundary was set as zero pressure condition, the pressure value in this figure was actually the pressure difference relative to the coronary outlet).**

8. For figure 1j C: Why are the cross-sections different between CFD and DL? You are using only the query points to predict the velocity, so how are the wall points affected?

Answer:

Thank you very much for your comments.

1. The shape of the cross-sections between CFD and DL should be the same. The reason for the difference is that in the post-processing process, we need to manually select and intercept the cross-section of the vessel. This leads to

the possibility that the cross-section shown may come from different locations. We re-cut the cross-section of the vessel carefully and ensured that it came from the same position of the model to the greatest extent (**Fig.1 Line647, Fig.1j C is shown here**). We apologize for our mistakes in this part of the work.

Fig.1j C

2. We use both the “query point cloud” and “model point cloud” to achieve hemodynamic prediction. To illustrate this, we have rewritten the method section (**Line374: Creation of datasets and proposal of deep learning network**), please check.

9. **For figure 1 in supplementary material: Please make the plot in logarithmic scale to easier to read.**

Answer:

We are very thankful for your comment. We refer to a large number of previous deep learning researches^{15,16,17,18}. When drawing the loss function curve, the abscissa uses real training times (Epoch) instead of “logarithmic scale”, even when the number of iterations is very large¹⁹. This descent way of the training curve can effectively represent the optimization process of the network. Therefore, we retain the training curve representation method in the original manuscript.

10. **For supplementary discussion: It seems that the method proposed by Liang et al. is performing in a similar fashion as yours. Why is your method better? Please provide more information and be specific on the problem setup of the other methods and the advantages of your method. Explain thoroughly.**

Answer:

We are very thankful for your comment.

At **Line71 (Main text)** “the 3D flow field model only appears in ideal geometry, and sample resolution in the dataset is too low to represent complex flow field distributions and geometric structures^{28,29}.”, their networks only accept the input with “prefix array-size”. This means, patient geometry should be normalized into “template”. Then, if the geometry cannot fit into that template, their network cannot accept that geometry. In other words, in case the geometry has just one

additional branch, they need to remake their network.

On the other hand, our network can predict the flow on any kind of geometry owing to the point cloud input. Even the number of point cloud (nodes) varies, our network can accept that unfixed input. Actually, our 1100 dataset had a different number of coronary arteries and different size of point cloud, but it works.

We added the following points into revised manuscript:

1. Supplementary information (**Line21**):

“Although Liang realizes the internal hemodynamic prediction of the ideal thoracic aortic model, the spatial resolution of its samples is still low, which could not accurately characterize the geometric characteristics of complex cardiovascular system. Liang's network only accepts the input data with prefix array-size. This means, patient geometry should be normalized into template (fixed number of meshes). Then, if the geometry cannot fit into that template, Liang's network cannot accept that input. Under the premise of more extensive information, our deep learning method uses limited data to achieve prediction accuracy similar to previous studies. However, our prediction objects are far more complex. Our network can predict the flow on any kind of geometry owing to using the point cloud format. Even the number of point cloud (nodes) varies, our network can accept that unfixed input. Actually, the 1100 models have different numbers of coronary arteries and different sizes of point clouds, but the network can still handle that. Combined with the universality analysis of the network, our deep learning method has many advantages.”

2. Supplementary information (**Line33**):

Supplementary Table 1 Comparison analysis of our deep learning method against previous studies

Network or method	Prediction output	Data set size	Input data format	Error function or accuracy
Our Deep Learning Method	3D Patient Personalized Cardiovascular Hemodynamics	1100	High resolution flexible point cloud	NMAE<6.5%, MRE<10%
Itu's Machine Learning Approach ¹	FFR Value	12,000	Geometric parameters	Accuracy=99.7%
Lee's Adversarial and Convolutional Neural Networks ²	2D Vortex Flow	500000	Grid cells with fixed number	32.8%<Error<1%
Guo's Deconvolution Network ³	3D Regular and Simple Flow	400000	Low resolution pixels with fixed number	MRE<3%
Liang's DNNs ⁴	3D Ideal Thoracic Aorta Hemodynamics	729	Low resolution meshes with fixed number	NMAE<6.5%

3. Discussion (Main Text, Line250):

“Liang et al. proposed a normalization method for deep learning²⁹. However, it was only suitable for large ideal aorta, not for the whole complex cardiovascular system, especially for small coronary branches. In addition, Liang et al. normalized the thoracic aortic models of different patients to the same meshes with only 80100 nodes. However, in this study, the mesh independence test (as seen in the Supplementary Methods) showed that for the thoracic aortic, when the number of meshes exceeded one million, the CFD simulation results can be considered to be stable. Therefore, the deep learning method proposed by Liang et al. still has great limitations in the application range and data resolution.”

11. It is confusing to me how come that when you use 100% of your dataset in the vortex region you get 28% error and when you are using 30% you are getting less. I believe that what you are trying to show is clear, but the method is confusing. Please explain more thoroughly and clearly your procedure.

Answer:

We are very thankful for your comment.

For Figure 4:

“100%” of size of the training set (x-axis) refers to 1000 models in the training set. The MRE value (y-axis) of “28%” at the 100% size of the training set refers to the error value obtained by only extracting the velocity of points on the vortex area for analysis. “30%” is the percentage of the number of points in the vortex area relative to the total number of query point clouds in one model. Here, vortex area was defined with Eigen Helicity method, Level 0.005, Actual Value 44.89 s⁻¹.

Combined with question 21 (about the points in the vortex region), first of all, we think that you have misunderstandings about the generation and extraction of point clouds. The point clouds are the connection points of CFD meshes (usually called nodes). CFD mesh generation is only related to the geometry of the model. Therefore, the distribution of point clouds depends only on the geometry of the model. The position of point cloud in the model is fixed, we don't change its spatial distribution. What we do is to directly extract and analyze the point cloud in a certain position (such as model wall, model interior; vortex area, laminar flow area; aorta area, coronary artery area, etc.) through the simulation software ANSYS.

We guess you mentioned about the confusion in Fig.4c, and according your comment of “30%”, the question may come up from the sentence “(original manuscript) Points in the aorta and superior aortic branch artery region accounted for more than 99% of the query point cloud, and more than 30% of the points were in the vortex region, ...” on Line 201. Clearly saying, to confirm the error

source of deep learning, we separated the region into two: vortex region and other region (including aorta, brachiocephalic, carotid and subclavian). Each error was calculated separately. Fig. 4c shows the error value of **only vortex region** when we increase the number of dataset.

We added some explanation on that sentence to make it clear as “**(Line201) Points in the aorta and superior aortic branch artery region accounted for more than 99% of the query point cloud, and more than 30% of the points in the whole region were located in the vortex region, which was the main source of prediction errors for the cardiovascular model. We extracted the points only in the vortex region which was defined with Eigen Helicity method, level 0.005, actual value 44.89 s⁻¹ for predicted results and calculated the error as shown in Fig. 4c.**”

- 12. It is also not clear how you make predictions. I understand that you train the same network on post and pre surgery data, it is not clear to me if then you provide a post and pre surgery geometry characteristics to the same network and make predictions. Please explain.**

Answer:

We are very thankful for your comment and we are sorry for our unclear description. As suggested, we clearly explained how to use the datasets to train the network and make predictions: “**(Line388) Based on this, the four groups—preoperative, postoperative, velocity, and pressure fields— of hemodynamic datasets were established, respectively. These four datasets need to be used as input to train the network independently. After that, we got four optimal network configurations to further predict the corresponding hemodynamics.**”

- 13. I understand that you enrich the data set by making random changes to the geometry characteristics, i.e. stenosis rate, but how does this affect the arterial geometry(i.e. the shape of the arterial walls)? Do you provide a different geometry or do you provide the same geometry, but with different characteristics? Please explain.**

Answer:

We are very thankful for your comment. “Geometric characteristics” have a direct influence on “geometry”. In the process of model expansion, when changing the “geometric characteristics (such as the diameter of the descending aorta),” the “geometry (such as the smoothness or shape of the descending aorta wall)” will also change. Therefore, we provide different “geometries” with different “geometric characteristics”.

In order to clearly show the geometric characteristics of the constructed models, we present two samples of original and expanded cardiovascular models from different patients, see **question 14**.

14. Could you provide some insight on how close is the test data geometry to the training set? Is it significantly different or close? This will help you show that the network can generalize to different architectures.

Answer:

We are very thankful for your comment. As suggested, we show in detail two sets of models (including original models and expanded models, training set and test set): **(Line355)** Based on this method, we extended one original model into nine new models, which meant that the total number of models increased to 1100, as shown in Supplementary Fig. 2.

Supplementary information **(Line37)**:

Model geometric parameters modification

In order to visually show the difference between the models in the training set and the test set, and to clearly show the modification of the model's geometric structure, we selected a model from the training set and the test set, and showed the modification results of its geometric structure, as shown in Supplementary Fig. 2.

Training set

Test set

Supplementary Fig. 2 Examples of models in the training and test sets. Examples of models in the training and test sets. A is the original model. B is the corresponding modified model. **a:** overall model; **b:** ascending aorta and aortic arch angulation; **c:** descending aorta; **d:** coronary artery details (LAD and LCX); **e:** stenosis. All model modifications follow the provisions of Table 3 in the main text.

Actually to the best of our knowledge, how to evaluate the similarity of complex artery “as quantitative value” is not established yet. After drastic progress of deep learning technique, current trend comes to “uncertainty” of deep learning results related to bias based on training data.

We also think that neural network only works correctly within “what he learned”. But unfortunately, the quantitative method to show “how much wider the network knows arterial geometry” is not established. Instead of that, we disclosed the range of modification and imposed random sampling and tried a number of attempts of network training (as Fig. 4c shows error bar) to neglect the influence of dataset sampling.

As we agree that your point is the issue for next generation, we added this point in the discussion as “**Regarding the datasets with several types of disease, we also need to establish the quantitative methodology to evaluate the variety of geometry as a training data.**” in **Line320**.

- 15. line 43: In my opinion, you should mention at this point the use of reduced order models in cardiovascular hemodynamics and especially in the case of predicting FFR. The reduced order models are inexpensive and accurate in some situations(see \textit{"Estimating the accuracy of a reduced-order model for the calculation of fractional flow reserve (FFR)."}). Why is it necessary to employ a 3D model of Navier-Stokes? What are the geometry characteristics that prohibit the use of reduced order models?**

Answer:

We are very thankful for your comment. We fully agree with you “the reduced order models are inexpensive and occur in some situations.” However, “reduced order models” is often highly targeted. For example, the reduced order model used to calculate FFR cannot calculate coronary blood flow. This limitation is also mentioned in the study of Liang et al.²⁰. Our deep learning method can predict 3D cardiovascular hemodynamics. For doctors and researchers, the results are very intuitive and easy to accept and understand. It is also very easy to calculate FFR and other parameters based on 3D hemodynamics. In the original Line 43, we want to emphasize the current shortcomings of CFD. Furthermore, the purpose of this research is to use deep learning instead of CFD. We do not focus on the calculation of a certain clinical parameter (such as FFR).

In addition, as mentioned in discussion chapter, **(Main Text, Line284)** “**Our deep learning method is highly universal, which is not limited to guiding the implementation of CABG and the treatment of CHD. It can analyze and reproduce the relationship between complex cardiovascular geometry and hemodynamics in a given dataset, which can be extended to the hemodynamic simulation of other organs and tissues, or even the flow field research under experimental conditions, such as replacing the steady flow 3D PIV experiment with sufficient data.**” This is also the reduced order models cannot achieve.

- 16. line 66-67: In my understanding you present these common CFD limitations to strengthen your argument about employing deep learning techniques for predicting the flow characteristics, but there are some points that you make that are not clear to me. For 2), if I understand correctly you are**

implementing your model to a steady state incompressible Navier-Stokes equation with constant boundary conditions, where the flow data are acquired by a conventional CFD simulator and a virtual surgery. This makes the flow pretty ideal, because the difference between ideal and patient measured flow is the presence of noise which makes the measurements not necessarily a solution to your system of equations. So, my first question is: What makes your set-up non-ideal and difficult to the point of proposing a new architecture and implement deep learning? Do you use both patient specific and data acquired by ANSYS or only data from ANSYS? Please elaborate.

Answer:

1. As mentioned in the discussion chapter, due to the lack of personalized boundary condition data in this study, we cannot carry out CFD simulation with personalized boundary conditions and propose a new deep learning architecture. **(Line297)** “The biggest limitation of this study is the lack of clinical data. In CFD simulation, there is no boundary condition information for patients. Currently we adopted constant values on inlets and outlets, which have been widely used among a number of geometries⁴⁴⁻⁴⁶. Therefore, the simulation results should include differences from real hemodynamics. On the other hand, this assumption may also facilitate the stability of prediction by moderating the variety of hemodynamics. In future approaches that include boundary conditions, another input channel will be required on the network. This input channel will take the patient's personalized boundary conditions as the input, and together with the model point cloud as the teaching signal to control the training process.”
2. We used cardiovascular models with patient specific geometries and uniform boundary conditions for ANSYS CFD simulation.

17. line 66-67: For 3): You claim that one point that makes your method superior in comparison with other methods is the use of a small dataset and other models need millions of samples. Please elaborate more on what you mean by small and large datasets. In your case, if I understand correctly you use 1100 geometries each consisting of 2 million points, which corresponds to million of samples. Do you mean that other studies require million of geometries?

Answer:

We are very thankful for your comment.

1. As suggested, we clearly show the amount of data required by different deep learning methods:

Supplementary information (Line33):

Supplementary Table 1 Comparison analysis of our deep learning method against previous studies

Network or method	Prediction output	Data set size	Input data format	Error function or accuracy
Our Deep Learning Method	3D Patient Personalized Cardiovascular Hemodynamics	1100	High resolution flexible point cloud	NMAE<6.5%, MRE<10%
Itu's Machine Learning Approach ¹	FFR Value	12,000	Geometric parameters	Accuracy=99.7%
Lee's Adversarial and Convolutional Neural Networks ²	2D Vortex Flow	500000	Grid cells with fixed number	32.8%<Error<1%
Guo's Deconvolution Network ³	3D Regular and Simple Flow	400000	Low resolution pixels with fixed number	MRE<3%
Liang's DNNs ⁴	3D Ideal Thoracic Aorta Hemodynamics	729	Low resolution meshes with fixed number	NMAE<6.5%

2. What you understand is correct. In all of these studies, all “samples” from the same “geometry” are called “one geometry”. For example, the research by Guo et al. requires 0.4 million “geometries”. We only need 1100 “geometries”.

18. line 66-67: Another point you should consider is the wall identification noise, which is a significant parameter of uncertainty in hemodynamics predictions in clinical practice, and how would this affect your method. Please provide some insight on that, because your method is relying on wall points from a mesh, which makes this a valid point.

Answer:

We are very thankful for your comment. This is the common limitation on image-based analysis including CFD. Berg exhibits²¹ the variability of vessel wall segmentation among 26 groups as “the Multiple Aneurysms Anatomy Challenge”. On that, morphology parameters varied up to 25%. This could lead an overestimation or underestimation of absolute value of hemodynamic parameters. But on the other hand, CFD result shows good consistency among the groups when STL file (already segmented geometry) was provided²² (Virtual Intracranial Stenting Challenge by Radaelli).

The ideal solution for that “wall identification noise” is an establishment of stable segmentation method or imaging method which can return a stable geometry.

Then, as more practical solution in the near future, normalization of parameters may help for stable use; such as “pressure-drop ratio against vessel diameter” or “relative WSS”.

We add the sentence on the discussion as:

(Line 326) “Thirdly, the uncertainty of vessel wall identification should be noted as the common limitation in image-based analysis including CFD. Present study

exhibits the flow estimation on the point clouds which generated on segmented blood vessel. Then, the flow field strongly depends on the quality of vessel wall segmentation. Though CFD results from the same STL file can exhibit good consistency among different research groups⁴⁹, still segmentation process from DICOM images can lead a variability of geometry⁵⁰. To overcome this uncertainty of real geometry, the establishment of stable segmentation method or normalization of hemodynamic parameter will be required.”

- 19. line 88-90: It is not obvious what you mean by "that even high-density point clouds can store a great deal of valid information with little data". Do you mean that you do not need to store the connectivity information? What make this method of storing to incorporate information in a sense that make this superior over other/ Is this just the memory capacity?**

Answer:

Thanks very much for your comment. We are sorry for our unclear description. What we want to express here in the manuscript is:

1. The high resolution CFD grid is irregular.
2. The point clouds used in this study are extracted from high-resolution CFD results, so their properties are as follows:
 - 1) Retain the high resolution of the mesh to the model geometry and flow field distribution;
 - 2) There is no connection information between the points (the nature of the point cloud itself).

This part corresponds to the low resolution characteristics of other formats (regularized grids, etc.) used in previous deep learning (**Line71** “the 3D flow field model only appears in ideal geometry, and sample resolution in the dataset is too low to represent complex flow field distributions and geometric structures^{28,29}”). It mainly emphasizes the high-resolution representation by point cloud. In order to avoid ambiguity, we revise this part as follows: (**Line89**) “Later, we converted the CFD results into high-density 3D point clouds. The point cloud inherited the ability of CFD results to characterize the geometric structure and flow field distribution of the model, which could characterize the complex flow field distribution and geometry of real cardiovascular models with high resolution^{35,36}. On this basis, preoperative and postoperative cardiovascular hemodynamic point datasets were established, respectively.”

As for the advantage description of point cloud compared with other formats, we have carried on the detailed analysis in the **Discussion** section. Please check it.

- 20. line 113: This is not correct. You cannot assess overfitting from the loss function value over epochs for the training set. Overfitting means that the model can make accurate predictions for the training set and not for the test set. If you provided a figure with the error on the test set during training then you probably mention overfitting.**

Answer:

We are very thankful for your comment and we totally agree with that. As suggested, we have corrected two sentences: **(Line116)** “The loss function fully converged.” **(Line148)** “The loss function converged faster.”

21. line 127: I believe that this is a misleading point, because you do not explain why you model cannot make accurate predictions in the region where the vortex is occurring and also you do not propose any improvement, you just show that if you assign more points in the region you can get better accuracy, which is not considered and improvement to you model. Also, \$28 \%\$ error for 2 million samples is not considered very accurate.

Answer:

We are very thankful for your comment. We think that you may have misunderstood the content of error analysis of vortex region (as pointed out in **question 11**).

We did not say anything about “assigning more points in the region we can get better accuracy.” We carried out control experiments to show that increasing the number of training models could improve the prediction accuracy of vortex region.

1. We compared the error difference between the vortex area and the overall model through the control experiment, and speculated that the reason for the high prediction error of the vortex part is due to its complex flow field distribution, as shown in **Fig. 4(Line680: Prediction error analysis. Using a model in the preoperative test set as a sample, a shows the vortex regions in its CFD simulation flow field, which are mostly distributed in the aorta and superior aortic branch artery rather than the grafts and coronary arteries. Compared to the entire model, the number of points in the vortex region of grafts and coronary arteries only accounted for 0.1% of all query points. For most models, there is an obvious vortex region in the ascending aorta area circled by the red box. In order to clearly indicate the distribution of points with high prediction errors (MRE> 10%), b shows the points with high errors in deep learning predicted velocity field of the same model as a (the circled area). The distribution of these points is highly consistent with the vortex region in a, which proves that the error mainly comes from the vortex region of the aorta and superior aortic branch artery. c shows the prediction errors along to the different size of training set only in the vortex region.)**.
2. We can't “assign more points in the region”. We can only directly extract the point cloud in the vortex region based on the results of CFD mesh generation. We clearly explained this in **Line379**: “*Pi* was the connection point of CFD meshes (usually called node). CFD meshes generation was only related to the geometry of the model. Therefore, the distribution of points depended only on the geometry of the model. The position of points in the model was fixed, we

could not change its spatial distribution. What we could do is to directly extract and analyze the points in a certain position through the simulation software.”

3. As shown in **Fig. 4(Line 689: c** shows the prediction errors along to the different size of training set only in the vortex region. **c** shows the influence of the training set size on MRE of the vortex region. We fix the test set and increase the size of the training set from 10% to 100%. Then we calculate and observe the MRE of the vortex region. When the size of the training set reach 100%, MRE still displays a significant downward trend. The minimum value of 28.53% is still far greater than the aorta and superior aortic branch artery (aorta region) MRE of $9.74 \pm 3.83\%$ as shown in Table 1. It further confirms the source of the prediction error and shows that it is necessary to increase the size of training sets.), the error reduction in the vortex region is due to an increase in the total number of models we use for training, from 10% (100 models) to 100% (1000 models). This proves that the increase in the number of models used for training is an effective means to improve the accuracy of prediction, and this method is also used in other deep learning research^{23,24}.
4. 28% is not the error of 2 million samples. 28% refers to the error of only the vortex region (30% of 2 million points, about 600000 points). The total error of 2 million points is shown in **Line 706 Table 1**($<10\%$).

22. line 141: I believe that you are referring to pressure being a scalar field. Please be a little bit more descriptive, because this might not be clear to the reader.

Answer:

We are very thankful for your comment. We totally agree with your comment. As suggested, we have corrected these sentences: **(Line144)** “Different from the velocity, the pressure in the flow field was scalar, that was, the pressure at a point had the same value in all directions. There were different vector components of velocity vector in X, Y and Z directions. Therefore, the pressure datasets as the network input contained less information than the velocity field datasets, which was reflected in the convergence speed of the loss function value versus epochs (as seen in Supplementary Fig. 1). The loss function converged faster.”

23. line 144: See above.

Answer:

Please check the answer of **question 22**.

24. It is not clear to me how can the two pressure distributions be close to each other granted that for incompressible Navier-Stokes you can only predict the

absolute pressure up to a constant. Can you provide some insight regarding that?

Answer:

We are very thankful for your comment. As we answered to your question 7, CFD provides the pressure distribution as a difference from outlet ($P=0$). In present study, we assign several settings: incompressible, laminar, iso-thermal, Newtonian, steady constant inlet and outlet, constant fluid properties... Then, finally based on the NS equations and equation of continuity, the last material to generate a flow field is only geometry and mesh quality in our CFD series. As for mesh quality, we did mesh-independence test and confirm the mesh used can have enough quality to resolve aorta hemodynamics.

Now our network learns the relationship between geometry and flow field, like well-skilled flow-engineer can imagine the vortex in curved pipe. As written in **Line 96**, “By extracting and integrating global and local features of the point cloud, the network could analyze and reproduce the relationship between vessel geometry in the point cloud datasets and the corresponding hemodynamics.”

- 25. It is not clear to me if the Bland-Altman analysis proves your point of the data not having any systemic bias. In figure 3d, you can see that there is in fact some bias, if I understand correctly your plot, which is not very small considering the flow order of magnitude (around 10^4).**

Answer:

We are very thankful for your comment and we totally agree it. Here, we just wanted to compare the consistency of the two methods when using CFD as the standard. Therefore, we amend it as follows: **(Line188)** “Also, the Bland-Altman analysis result is as shown in Fig. 3c and 3d: 97 sets of FFR data and 97 sets of improved flow data fall within the 95% confidence interval (FFR: -0.07780-0.09254; Flow: -1.282-0.8568), which confirmed that the clinical indicators calculated by these methods were in agreement.”

- 26. line 208: Could you provide more information regarding these cases of vortex prediction that you are referring to? What was the setup? How was it similar or different than yours?**

Answer:

We are very thankful for your comment. As suggested, we explain more details here: “**(Line 212)** Taking Lee's research as an example³⁹, in a 2D plane with a size of 250×250 (grid cells), 500,000 vortex samples were needed to train the network. The number of samples was far more than that of this study. However, the complexity of the vortex (2D) was lower than that of this study (3D).”

27. line 238: Please explain where do these numbers come from. Why $800 \times 1200 \times 950$?

Answer:

We are sorry for the unclear description. As suggested, we modify it: **(Line244)** For example, when using Guo et al's normalization method to process the cardiovascular models in this study²⁸, we should place the model in a three-dimensional space cuboid. The length, width and height of the cuboid should be the maximum value in the corresponding direction of the model. For a cardiovascular model with a length of 80 mm, a width of 120 mm and a height of 95 mm, when using 0.1 mm as the segmentation size, one 3D cardiovascular model needs to have $800 \times 1200 \times 950 = 836,000,000$ normalized points to contain the mean shape, which is unacceptable for network and GPU processors.”

28. line 252-254: Please explain what you mean by spatial disorder. The readers might come from a diverse background, so this might not be obvious to them.

Answer:

We are very thankful for your comment. As suggested, we explain more details here: **(Line271)** “Since there is no connectivity information between the points, there is no specific input sequence for the points. That is to say, when N points are used as input to the network, due to the different input sequence, there may be N! input permutations, that is, the disorder of the point cloud.”

29. line 258: "local feature... teacher's signal" I not sure what you mean at this point. Please rephrase or elaborate more.

Answer:

We are sorry for our unclear description. As suggested, we rephrase it: **(Line277)** “Global features provide the outer geometry information of the model, which can help the query point cloud to obtain its position inside the model. Under the uniform CFD boundary conditions, the position of the query point is corresponding to the flow field. The local flow field characteristics and the corresponding position information can be used as teacher signals to help the network learn the hemodynamic values of a specific position. In this way, the spatial relationship is effectively introduced to help the network attain correspondence between the model geometry and the flow field distribution.”

30. line 386-387: What do you mean by local ERR values? Could you elaborate more?

Answer:

We are sorry for our unclear description. As suggested, we explain more details: **(Line467)** “In order to avoid the impact of this variation on the evaluation results, local ERR (The model was divided into several parts, and the ERR value of a certain part, such as the left anterior descending branch, was called local ERR.) values were obtained to assess the prediction accuracy on small-to-large parts.”

- 31. line 391: Can there be multiple stenosis in your LAD branch? Is this a part of the data set? Please be more specific about your dataset creation because it might be confusing to the reader.**

Answer:

We are very thankful for your comment.

1. Yes, there are some models with multiple stenosis in the LAD branch.
2. We show the detail of how to create our dataset, please refer to the answer of **question 14 (Test set d and e)**.

Grammar/Typos::

- 1. line 36: ...obtaining hemodynamics. Please rephrase because it is not clear what you mean. Hemodynamics is a very general term.**

Answer:

We are very thankful for your comment. We are sorry for our unclear expression. As suggested, we rephrase “hemodynamics” to “hemodynamic parameters including velocity and pressure” in the revised sentence **(Line40)**. Please check it.

- 2. line 41: ...hemodynamic numerical models. Please rephrase this is not correct grammar.**

Answer:

We are very thankful for your comment. We are sorry for our wrong expression. As suggested, we rephrase “hemodynamic numerical models” to “velocity and pressure” in the revised sentence **(Line45)**. Please check it.

- 3. line 55: ...prediction. I believe that a more accurate term would be regression.**

Answer:

We are very thankful for your comment. As suggested, we rephrase “prediction”

to “regression” in the revised sentence (Line62). Please check it.

4. **line 56: ... the computing power of deep learning. It is not clear to what you are referring to and what you mean by the term computing power.**

Answer:

We are very thankful for your comment. We are sorry for our unclear expression. As suggested, we rephrase “... the computing power of deep learning” to “Advanced deep learning algorithm” in the revised sentence (Line62). Please check it.

5. **line 57: ...expansion. I believe that this is not a proper term to describe what you mean. Please correct the language.**

Answer:

We are very thankful for your comment. And we are sorry for this wrong expression. As suggested, we rephrase “Due to the development and expansion of deep learning techniques,” to “Due to the development of deep learning techniques,” in the revised sentence (Line64). Please check it.

6. **line 76: "Based on the above background": Please rephrase.**

Answer:

We are very thankful for your comment. As suggested, we rephrase “Based on the above background” to “In this study” in the revised sentence (Line82). Please check it.

7. **line 78: "at once": Please rephrase.**

Answer:

We are very thankful for your comment. As suggested, we removed “at once” in the revised sentence (Line85). Please check it.

8. **line 79: "velocity and pressure field\textbf{s}”.**

Answer:

We are very thankful for your comment. As suggested, we rephrase “which could reproduce velocity and pressure field from geometrical features” to “which could

predict the velocity field and pressure field based on the geometric features of the model” in the revised sentence **(Line84)**. Please check it.

9. line 81-82: "simulation implementation". Please rephrase.

Answer:

We are very thankful for your comment. As suggested, we rephrase “CABG simulation implementation” to “simulation of CABG surgery” in the revised sentence **(Line87)**. Please check it.

10. line 82: “the CFD method”. Change to “a CFD method”.

Answer:

We are very thankful for your comment. As suggested, we rephrase “the CFD method” to “a CFD method” in the revised sentence **(Line88)**. Please check it.

11. line 89: “even high-density...valid information”. Probably you need to remove even and valid.

Answer:

We are very thankful for your comment. We are sorry for our wrong expression. As suggested, we rewrote this part. Please check the answer to question 19 (Technical Comments, Clarifications and Suggestions)

12. line 94: "optimal weight assignment": Please rephrase.

Answer:

We are very thankful for your comment. As suggested, we rephrase “The deep learning network only needs to complete the optimal weight assignment once.” to “The deep learning network only needs to be trained once.” in the revised sentence **(Line98)**. Please check it.

13. line 98: "we defined.." Please change this to "we define”.

Answer:

We are very thankful for your comment. We are sorry for our unclear expression. As suggested, we rephrase “defined” to “define” in the revised sentence **(Line101)**. Please check it.

14. line 112: "learning curve": You should probably rephrase that to "Loss function value versus epochs".

Answer:

We are very thankful for your comment. As suggested, we rephrase "learning curve" to "Loss function value versus epochs" in the revised sentence (Line115 and Line148). Please check it.

15. line 116-117: "It was confirmed ..." Please rephrase.

Answer:

We are very thankful for your comment. As suggested, we rephrase "It was confirmed ..." to "It showed..." in the revised sentence (Line119). Please check it.

16. line 124: "mainly due to the part". Please rephrase.

Answer:

We are very thankful for your comment. We are sorry for our wrong expression. As suggested, we rephrase "This was mainly due in part to the larger magnitude of flow" to "This was mainly due to the larger magnitude of flow" in the revised sentence (Line127). Please check it.

17. line 162: "in the test set was input". Please rephrase.

Answer:

We are very thankful for your comment. We are sorry for our unclear expression. As suggested, we rephrase "in the test set was input" to "the point coordinate space information of the cardiovascular model in the test set was input to the network" in the revised sentence (Line166). Please check it.

18. line 172: "had broad application..". Please change to "has ...".

Answer:

We are very thankful for your comment. As suggested, we changed it in the revised sentence: (Line176) "...deep learning has ..." Please check it.

19. line 195-196: Please rephrase. The grammar is not proper.

Answer:

We are very thankful for your comment. We are sorry for our wrong expression. As suggested, we rephrase “We verified the points of the entire cardiovascular model with larger error function values (MRE>10%), which were highly coincident with the vortex region in the CFD calculation results, as shown in Fig. 4a and 4b.” to “We extracted regions with large prediction error function values (MRE>10%) in the entire cardiovascular model. These regions were highly consistent with the vortex regions in the CFD calculation results, as shown in Fig. 4a and 4b.” in the revised sentence (**Line198**). Please check it.

20. line 214: "dataset is very necessary". Remove "very".

Answer:

We are very thankful for your comment. As suggested, we removed “very” in the revised sentence: (**Line221**) “...dataset is necessary ...” Please check it.

Reviewer#2

Remarks to the Author:

Summary:

The paper addresses the challenge of non-invasively predicting hemodynamic parameters from imaging data. Clinically approved methods for this purpose involve computational fluid dynamics, which bear limitations such as extensive computation times and sensitivity to boundary conditions. This paper follows the recently emerging approach of approximating simulations with deep learning models. For this purpose, authors propose a novel network architecture which builds upon the highly recognized PointNet architecture. PointNet is capable of directly processing point clouds, which correspond to the input and output format of simulations.

The sparse nature of point clouds, enables direct processing of the simulation input data corresponding to the whole system of aorta and coronaries without compression. The model is trained and evaluated on point clouds extracted from in-vivo image data.

Evaluation shows good agreement of model output with the simulated prediction targets, in-line with performance of previous work.

Thank you very much for your high evaluation and comments on our research. Based on your comments, we have made the following amendments to the manuscript and explained in detail in the answers to specific questions below. Please check it.

General Comment:

The paper is well written and structured with a few spelling and grammar mistakes (see minor issues). The presented results do not show any methodological flaws.

However, clarity of the motivation and discussion of the deep learning architecture needs improvement. At the current stage it is cumbersome to assess which parts of the proposed architecture are novel.

While results and motivation support the claim of superiority of PointNet-based architectures, unclarities remain in the motivation of the architecture.

Answer

Thank you very much for your comments. As suggested, we rewrite the method part to clearly explain the design principles and innovation of our network structure. At the same time, we add an ablation experiment (compare the prediction error between our network and the original PointNet) and a control experiment (with or without “shared weight”) to further prove the superiority of our network structure. Please refer to the answers to questions 1 and 2 (**Line392-436**).

Another main weakness lies in missing evaluation on in-vivo measured hemodynamic parameters as only simulated parameters are used for training and testing throughout this work.

Answer

Thank you very much for your comments. Based on your comments, we made the following changes in the manuscript:

1. Clarify the main research purposes in **Introduction**:
 - 1) This study aims to develop a deep learning method to realize the fast prediction of velocity and pressure field of cardiovascular system to reduce the high computational cost of CFD.
 - 2) This study aims to develop a deep learning approach that can realize 3D personalized cardiovascular system hemodynamic prediction, in view of the fact that previous deep learning methods are only suitable for 2D or ideal models.
2. Clarify the limitations of the current study in **Discussion**:

Limited by the patient's clinical data, this study lacked information on the flow/pressure measured by the catheter. We only compared the results of CFD and deep learning. We give a detailed statement on this point in the **limitations section of the discussion chapter (Line305)** “In the analysis of prediction accuracy, we only compared the prediction results of deep learning with CFD, but lack of comparison with clinical measured data of patients (such as invasive FFR). Itu et al.⁴⁷ and Tesche et al.⁴⁸ proved that under the premise of

good consistency between the FFR calculated by deep learning and CFD, compared with the invasive FFR, the FFR values calculated by these three methods were also with good consistency, which is also one of the further work we need to accomplish.”

Originality and significance:

Application of a PointNet-based algorithm to hemodynamic simulations has not been done before to the best of my knowledge. This application is interesting because it enables direct application of the model as a surrogate for computational fluid dynamics.

The sparsity of point clouds enables the given approach to process the entirety of information associated with the combined system of aorta and coronary arteries. Thorough evaluation regarding the origin of most erroneous predictions supports the claim of general applicability of the method to other prediction scenarios involving simulations.

However, shortcomings in the evaluation of the presented network architecture prevent conclusions about utility of the presented architectural novelties with respect to the original PointNet.

Answer

Thank you very much for your comments. As for the **design principle** of network structure, the **innovation point** of network, the **difference** between network and original PointNet, we give a detailed improvement scheme in **question 1 and 2**. Please refer to the **answers to questions 1 and 2**.

Major Issues:

- 1. Changes made to the original PointNet architecture were not explicitly stated. While the methods section (in the end of the paper) enables identifying these changes, they should be stated explicitly. The original PointNet architecture also incorporates local and global features, however both derived from the query points. Please motivate the separate model branch considering this. More severely, these changes were not explicitly evaluated. To strengthen justification of the model branch, I propose an ablation experiment where the original PointNet is directly applied. Furthermore, shared weights impose a limitation to the expressivity of the network. It would be interesting to evaluate beneficiality of shared weights in the first two layers by comparing results against a training run without shared weights. Current absence of these evaluations makes some of the claims in the discussion seem unsupported, i.e. lines 255 - 260.**

Answer:

Thank you very much for your comment. As suggested, we rewrote this part.

1. Clearly explain the similarities and differences between our network and the original point network.
2. “An ablation experiment” is added to compare the prediction error between our network and the original PointNet.
3. A control experiment is added to explain the importance of "sharing weight".

The details are as follows: **(Line395)** “The segmentation network structure of PointNet³⁷ could realize feature extraction and hemodynamic prediction of point cloud. This study inherited the concepts of global feature and local feature proposed by the original PointNet, and optimized the network structure. Since the original PointNet only had one single input channel, global features and local features were extracted from the same and all input points, which could help the PointNet identify the relationship between these two features. However, it was inevitable that there would be duplication between the two features, and then some effective and specific information would be lost. In order to solve this problem, a network structure with double input and double sampling channels was proposed in this study. The structure and parameter setting were shown in Fig. 5. For each model in the dataset, we extracted two types of point clouds. One was the model point cloud, which only included spatial coordinates for the outermost points of the cardiovascular model. These points could represent the global features of the overall model geometry. The second was the query point cloud, which included the remaining points inside the cardiovascular model. These points contained local features such as the spatial coordinates of each point and its corresponding hemodynamics. The 3D deep learning network had two independent input channels that corresponded to these two point clouds. Two feature extraction parts were directly connected to their respective input and sampling channels. This effectively enhanced the extraction of effective and specific information from these two features, and improved the prediction accuracy. For the same point cloud data, the comparison of prediction MRE from the original single channel PointNet and our dual channel network was shown in Table 4. The prediction ERR of our dual channel network was significantly lower than that of the original PointNet.

Global features were the global geometric information of the model. Local features referred to the distribution of each point and the corresponding flow field distribution inside the model. These two features both contained the geometric features of the same cardiovascular model, namely commonality. The two features also had different effective information, namely difference. The network needed to extract commonality and difference and learned the correlation between them to further realize the flow field prediction.

Based on the above principle, the network construction scheme was as follows:

To enhance the commonality and correlation, the first two feedforward fully-connected layers (FC1 and FC2) of the two feature extraction sections shared weights, which meant they shared the same underlying feature extraction methods. In order to evaluate the necessity of sharing weights, we compared the

network without shared weights with the results of this study, as shown in Table 5. The results showed that the shared weight could effectively reduce the prediction error. The two feature extraction sections also had independent feedforward fully-connected layers (FC3 and FC4), which further enhanced the ability of the network to recognize the effective and specific information (difference) of global features and local features. After FC3 and FC4, the global and local features contained in the two point clouds were represented as a $N * 512$ and $N * 128$ -dimensional vectors, respectively. We stitched together the two vectors to form an $N * 640$ -dimensional vector. This vector contained both the global features of the model point cloud and the local features of the query point cloud, which helped the network further integrate the correlation between the two features. The last part of the network was feedforward fully-connected layers (FC5 and FC6), which were used to yield hemodynamic results.

(Line713)

Table 4. Comparison of MRE from original PointNet and our network *

Hemodynamic	Velocity field	Pressure field
Original PointNet	18.42±6.71	14.59±5.31
Our Network	9.77±3.86	7.61±1.99

* The MRE is calculated according to the hemodynamic prediction values of preoperative aorta and superior aortic branch artery.

Table 5. Comparison of MRE from network with or without shared weights*

Hemodynamic	Velocity field	Pressure field
Without shared weights	16.37±5.43	13.42±5.21
With shared weights	9.77±3.86	7.61±1.99

* The MRE is calculated according to the hemodynamic prediction values of preoperative aorta and superior aortic branch artery.”

2. While utility of global (point 1) and local (point 2) features is intuitive, point 3 in the desired functions of the algorithm needs further clarification.

336: "The ability to learn and reproduce the relationship between two features, especially regarding commonality, difference, and correlation."

Does "two features" refer to local vs global features? Please briefly explain why commonality, difference, and correlation is a required learning goal.

Answer:

Thank you very much for your comments. We are sorry for the unclear

description in this part.

1. The “two features” refer to global feature and local feature.
2. Global features are the global geometric information of the model. Local features refer to the distribution of each spatial point and the specific flow field distribution characteristics in the model. These two features both contain the geometric features of the model, namely “commonality”. The two features also have different effective information, namely “difference”. The network needs to extract “commonality (geometric structure)” and “difference (internal flow field distribution)” to learn the "correlation" between them to further realize the flow field prediction. This is the basic principle of network design. The details of it was explained following.

As suggested, we rewrote this part. The details are as follows: **(Line416)** “Global features were the global geometric information of the model. Local features referred to the distribution of each point and the corresponding flow field distribution inside the model. These two features both contained the geometric features of the same cardiovascular model, namely commonality. The two features also had different effective information, namely difference. The network needed to extract commonality and difference and learned the correlation between them to further realize the flow field prediction.

Based on the above principle, the network construction scheme was as follows:

To enhance the commonality and correlation, the first two feedforward fully-connected layers (FC1 and FC2) of the two feature extraction sections shared weights, which meant they shared the same underlying feature extraction methods. In order to evaluate the necessity of sharing weights, we compared the network without shared weights with the results of this study, as shown in Table 5. The results showed that the shared weight could effectively reduce the prediction error. The two feature extraction sections also had independent feedforward fully-connected layers (FC3 and FC4), which further enhanced the ability of the network to recognize the effective and specific information (difference) of global features and local features. After FC3 and FC4, the global and local features contained in the two point clouds were represented as a $N * 512$ and $N * 128$ -dimensional vectors, respectively. We stitched together the two vectors to form an $N * 640$ -dimensional vector. This vector contained both the global features of the model point cloud and the local features of the query point cloud, which helped the network further integrate the correlation between the two features. The last part of the network was feedforward fully-connected layers (FC5 and FC6), which were used to yield hemodynamic results.”

3. **"Concerning the dataset, each sample must have enough spatial resolution to resolve complex flow field and model geometry. And concerning the network, enhancing its feature extraction capabilities would be necessary so that it could efficiently obtain valid information with a limited sample number."**

Please specify what exactly you mean by enhanced feature extraction capability. This should be picked up in the methods section. How do you

achieve what you mean by enhanced feature extraction capabilities?.

Answer:

Thank you very much for your comments, which helps us to make the logic of the paper correct and more rigorous.

1. We rewrite the method part. We explained the design principle of the network in detail (dual sampling channel, shared weight layer, etc.), and added “an ablation experiment” and “a control experiment” to show that our network has stronger feature extraction ability and prediction accuracy than other network structures (such as the original PointNet or the network without shared weight layers). Please refer to the answers to questions 1 and 2.
2. However, due to the difference of data format and network structure, we cannot input the same training samples into other networks except PointNet to compare the prediction accuracy, and then illustrate the strength of the network feature extraction ability. Moreover, as for “a limited sample number”, this limitation is not obvious because 729 samples of ideal thoracic aortic model were used in the latest study of Liang et al²⁰. What we want to emphasize here is that there must be a new network structure corresponding to the new data format. Therefore, the description of “strong feature extraction ability” is not suitable here. In order to avoid ambiguity, we change this part into: **(Line69)** “The main limitations of these studies are: 1) most studies focus on 2D flow fields, which have a limited scope of application³¹⁻³⁴, 2) the 3D flow field model only appears in ideal geometry, and sample resolution in the dataset is too low to represent complex flow field distributions and geometric structures^{28,29}. For CABG surgery, a cardiovascular model with small grafts and coronary branches has an intricate geometry and internal flow field distribution. Therefore, in this study, in order to accurately predict complex 3D cardiovascular hemodynamics with limited samples, new requirements to adapt a flexibility and high resolution on the input geometry have been imposed on datasets and deep learning networks, which is also the main technical problem and contribution of this study. Concerning the dataset, each sample must have enough spatial resolution to resolve complex flow field and model geometry. Therefore, it is necessary to find a new, high-resolution sample representation format. And concerning the network, it is necessary to propose a new network structure that can effectively handle the new sample format.”

4. The following two statements seem to contradict each other.

1. (line 43) "When CFD is used to calculate the hemodynamics of a complex cardiovascular model with small grafts and coronary branches, even high-performance computing clusters will usually need several hours of iteration to ensure model accuracy."

2. (line 163) "For the CFD method, the calculation time of one model on an Intel Xeon Gold 6148 2.4Ghz × 2 CPU server was about 10 minutes."

Please elaborate.

Answer:

Thanks very much for your comments. We are sorry for the unclear description. According to the suggestion, we explain in detail the computational time of CFD under different boundary conditions: **(Line47) "When subjects' personalized CFD boundary conditions (e.g., the inlet is set to pulsatile flow, and the outlet pressure is an invasive measured value) are used to calculate the hemodynamics of complex cardiovascular models with small grafts and coronary branches, even high-performance computational clusters usually require several hours of iteration to ensure the accuracy of the model. Even with ideal boundary conditions (e.g., steady flow at the inlet and zero pressure at the outlet), CFD method also requires calculation time about ten minutes."**

- 5. "Compared to other 3D data formats (e.g., voxel grids), the point cloud format has a simple and unified structure. It does not introduce irregular shape and connection information, which means that even high-density point clouds can store a great deal of valid information with little data"**

Please explicitly establish the link between data irregularity and compressibility.

Answer:

Thanks very much for your comment. We are sorry for our unclear description. What we want to express here in the manuscript is:

1. To resolve the complex shape of cardiovascular system with the size range of 1mm(coronary)-30mm(aorta), unstructured mesh with a diameter-dependent grid size is suitable. Then, our CFD results appears as inhomogeneous nodes in spatial.
2. The point clouds used in this study are extracted from high-resolution CFD results, so their properties are as follows:
 - 1) Retain the high resolution of the mesh even in small artery to the model geometry and flow field distribution;
 - 2) There is no connection information between the points (the nature of the point cloud itself).

This part corresponds to the low resolution characteristics of other formats (regularized grids, etc.) used in previous deep learning **(Line71 "the 3D flow field model only appears in ideal geometry, and sample resolution in the dataset is too low to represent complex flow field distributions and geometric structures")**. It mainly emphasizes the high-resolution representation by point cloud. In order to avoid ambiguity, we revise this part as follows: **(Line89) "Later, we converted the CFD results into high-density 3D point clouds. The point cloud inherited the ability of CFD results to characterize the geometric structure and flow field**

distribution of the model, which could characterize the complex flow field distribution and geometry of real cardiovascular models with high resolution^{35,36}. On this basis, preoperative and postoperative cardiovascular hemodynamic point datasets were established, respectively.”

As for the advantage description of point cloud compared with other formats, we have carried on the detailed analysis in the discussion section **(Line 230-296)**. Please check it.

- 6. Limitations of this work with respect to previous work seems to be missing a key aspect. While a model operating on point-clouds is ideal for replicating simulation results, in-vivo data is structured on a grid and discarding this structure in favor of a point cloud erases potentially relevant relational information. Please elaborate on this potential limitation.**

Answer:

Thanks very much for your comments. As suggested, we state this potential limitation: **(Line312)** “In addition, the point cloud data used in this study is extracted from the CFD meshing result. In the point cloud extraction process, we deleted the connection relationship between the grids. Although the point cloud can reproduce the CFD flow field prediction results, it also brings potential limitations, such as the loss of correlation information between different nodes in the original CFD results and the introduction of the disorder of point clouds.”

- 7. 21 (Abstract): "Our deep learning method is significantly better than existing deep learning approaches..." Please be more specific on what is better, e.g. accuracy, computation time, applicable regions. The comparison to previous work in the supplement does not support this claim, as similarly well performing methods are listed (i.e. Liang 2020).**

Answer:

Thanks very much for your comments. As suggested, we have made the following modifications:

In the **Abstract** part, specific description is added: **(Line18)** “The statistical analysis shows that the hemodynamic prediction results of deep learning are in agreement with the conventional CFD method, but the calculation time is significantly reduced 600-fold. In terms of sample resolution over 2 million of nodes, prediction accuracy of around 90%, computational efficiency to predict cardiovascular hemodynamics within 1 second, and universality for applying complex arterial system, our deep learning method can meet the needs of most situations.”

- 8. 112: "...the learning curve was made available (as seen in Supplementary Fig. 1). The loss function fully converged without overfitting." Does this learning curve correspond to evaluation on the training or on a separate validation set? Convergence on training data does not enable outruling overfitting, which typically only shows up on the validation set.**

Answer:

We are very thankful for your comment and we totally agree with that.

1. This learning curve corresponds to the evaluation of the training set. We do not plot the validation loss during training. In this study, the test set is completely separated from the training data. We verify the universality of our deep learning method by obtaining high prediction accuracy on a completely independent test set.
2. As suggested, we have corrected these two sentences: **(Line116)** “The loss function fully converged.” **(Line148)** “The loss function converged faster.”

Minor Issues:

1. **364: We used the mean absolute error as the regression loss function, which made the network more robust to outliers in the input data. Please explicitly state the alternative.**

Answer:

We are very thankful for your comment. We are sorry for the wrong expression here.

1. Previous studies using deep learning to predict the flow field used mean absolute error as the loss function to prove that mean absolute error is efficient and reliable^{20,25}, which is why we chose it.
 2. There is no alternative to loss function. We are very sorry for the confusion caused by our misunderstanding of “robustness” here.
We have revised the manuscript as follows:
(Line439) remove the error description “**which made the network more robust to outliers in the input data.**”
(Line137) here, we want to show that our network can accurately predict the velocity or pressure field of the corresponding model regardless of whether the "graft" exists or not. We sum it up as “high performance”, that is, “**The proposed network could effectively identify significant and non-significant disturbances of the graft on the flow field, which highlighted its high performance.**”
2. **"The preoperative and postoperative datasets needed to be separately trained as inputs for the network. During the training process, we saved the optimal network parameter configurations for both training sets."**
This explanation seems to imply that two separate models were trained. Please state if this was the case or not.

Answer:

We are very thankful for your comment and we are sorry for our unclear description.

1. Yes, the datasets in this study need to be used as input to train the network independently.
2. As suggested, we clearly explained how to use the datasets to train the network and make predictions: **“(Line388) Based on this, the four groups—preoperative, postoperative, velocity, and pressure fields— of hemodynamic datasets were established, respectively. These four datasets need to be used as input to train the network independently. After that, we got four optimal network configurations to further predict the corresponding hemodynamics.”**

3. **59: "For example, Guo et al. put forward a calculation method of flow around simple geometric models based on convolutional neural networks. And Liang et al. proposed a deep learning method to predict simplified thoracic aortic hemodynamics."**

It is not clear why these two examples were chosen from amongst the references that are subsequently discussed regarding their limitations.

Answer:

We are very thankful for your comment. We apologize for the unclear statement in this part.

This part is a review of the research on using deep learning to predict flow fields. Guo's research focuses on 2D flow fields prediction. Liang's research is aimed at 3D flow fields prediction. They are representative. As suggested, we added an explanation here: **“(Line65) For example, Guo et al. put forward a calculation method of 2D flow around simple geometric models based on convolutional neural networks²⁸. And Liang et al. proposed a deep learning method to predict 3D simplified thoracic aortic hemodynamics²⁹.”**

4. **Table 1. The headline "The error functions of the velocity field" gives the impression that the table describes the analytic function itself, while it actually presents evaluations of this function. I would propose replacing "functions" with "metrics" or "Performance evaluation of the velocity field".**

Answer:

We are very thankful for your comment. As suggested, we changed it in the revised manuscript: **(Line706) “Table 1. Performance evaluation of the velocity field”** and **“Table 2. Performance evaluation of the pressure field”** Please check it.

5. 377: Please discuss suitability of both loss functions for this specific task.

Answer:

We are very thankful for your comment. As suggested, we further explained the "suitability" of the error function for this study. **“(Line458) NMAE can characterize the error of the deep learning prediction result relative to the true value of the overall flow field (CFD result). MRE can characterize the error of the deep learning prediction value relative to the true value at all query points of the model. The definition of the error function draws on previous studies. The comparative analysis results are shown in Supplementary Table 1. In this study, ERR is designed to evaluate the velocity or pressure fields represented by point clouds. For other parameters (such as FFR calculated by pressure field, etc.), new ERR should be defined according to the specific situation.”**

6. 391: “...the highest stenosis rate was selected.”  “... the stenosis with highest degree was selected.”.

Answer:

We are very thankful for your comment. As suggested, we changed it in the revised sentence: **(Line474) “...the stenosis with highest degree was selected.”** Please check it.

7. 148: “...pressure distribution in of the...”  “...pressure distribution of the...”.

Answer:

We are very thankful for your comment. As suggested, we changed it in the revised sentence: **(Line152) “...the pressure distribution of the ...”** Please check it.

8. 166: “..., this process only need to be completed...”  “..., this process only needs to be completed...”.

Answer:

We are very thankful for your comment. As suggested, we changed it in the revised sentence: **(Line170) “...process only needs to be...”** Please check it.

9. 206: “...much more training set data than ours, ...”  “...much more training data than ours, ...”.

Answer:

We are very thankful for your comment. As suggested, we changed it in the revised sentence: **(Line211)** "...to predict complex vortexes required **much more training data** than ours, ..." Please check it.

10. 244: "...generally called as node."  "...generally called a node.".

Answer:

We are very thankful for your comment. As suggested, we changed it in the revised sentence: **(Line261)** A point cloud is the connection point of CFD meshes and **is generally called a node**. Please check it.

References for response letter

1. Gold, J. P. *et al.* Improving outcomes in coronary surgery: the impact of echo-directed aortic cannulation and perioperative hemodynamic management in 500 patients. *Ann. Thorac. Surg.* **78**, 1579–1585 (2004).
2. Qiao, A. & Liu, Y. Influence of graft–host diameter ratio on the hemodynamics of CABG. *Biomed. Mater. Eng.* **16**, 189–201 (2006).
3. Rezaeimoghaddam, M. *et al.* Patient-specific hemodynamics of new coronary artery bypass configurations. *medRxiv* (2020).
4. Arefin, M. S. Hemodynamic and structural effects on bypass graft for different levels of stenosis using fluid structure interaction: A prospective analysis. *J. Vasc. Nurs.* **37**, 169–187 (2019).
5. Kitabata, H. *et al.* Incidence and predictors of lesion-specific ischemia by FFRCT: learnings from the international ADVANCE registry. *J. Cardiovasc. Comput. Tomogr.* **12**, 95–100 (2018).
6. Douglas, P. S. *et al.* 1-year outcomes of FFRCT-guided care in patients with suspected coronary disease: the PLATFORM study. *J. Am. Coll. Cardiol.* **68**, 435–445 (2016).
7. Patel, M. R. *et al.* 1-year impact on medical practice and clinical outcomes of FFRCT: the ADVANCE Registry. *JACC Cardiovasc. Imaging* **13**, 97–105 (2020).
8. Itu, L. *et al.* A machine-learning approach for computation of fractional flow reserve from coronary computed tomography. *J. Appl. Physiol.* **121**, 42–52 (2016).
9. Tesche, C. *et al.* Coronary CT angiography–derived fractional flow reserve: machine learning algorithm versus computational fluid dynamics modeling. *Radiology* **288**, 64–72 (2018).
10. Febina, J., Sikkandar, M. Y. & Sudharsan, N. M. Wall Shear Stress Estimation of Thoracic Aortic Aneurysm Using Computational Fluid Dynamics. *Comput. Math. Methods Med.* (2018). doi:10.1155/2018/7126532
11. Qi, C. R., Su, H., Mo, K. & Guibas, L. J. Pointnet: Deep learning on point sets for 3d classification and segmentation. in *Proceedings of the IEEE conference on computer vision and pattern recognition* 652–660 (2017).
12. Vignon-Clementel, I. E., Figueroa, C. A., Jansen, K. E. & Taylor, C. A. Outflow boundary conditions for 3D simulations of non-periodic blood flow and pressure fields in deformable arteries. *Comput. Methods Biomech. Biomed. Engin.* **13**, 625–640 (2010).
13. Grigioni, M. *et al.* A mathematical description of blood spiral flow in vessels: application to a numerical study of flow in arterial bending. *J. Biomech.* **38**, 1375–1386 (2005).
14. Migliori, S. *et al.* A framework for computational fluid dynamic analyses of patient-specific stented coronary arteries from optical coherence tomography images. *Med. Eng. Phys.* **47**, 105–116 (2017).
15. Lee, C. S., Baughman, D. M. & Lee, A. Y. Deep learning is effective for

- classifying normal versus age-related macular degeneration OCT images. *Ophthalmol. Retin.* **1**, 322–327 (2017).
16. Mao, Z., Yao, W. X. & Huang, Y. EEG-based biometric identification with deep learning. in *2017 8th International IEEE/EMBS Conference on Neural Engineering (NER)* 609–612 (IEEE, 2017).
 17. Chen, C. L. *et al.* Deep learning in label-free cell classification. *Sci. Rep.* **6**, 21471 (2016).
 18. Lakhani, P. & Sundaram, B. Deep learning at chest radiography: automated classification of pulmonary tuberculosis by using convolutional neural networks. *Radiology* **284**, 574–582 (2017).
 19. Lee, S. & You, D. Data-driven prediction of unsteady flow over a circular cylinder using deep learning. *J. Fluid Mech.* **879**, 217–254 (2019).
 20. Liang, L., Mao, W. & Sun, W. A feasibility study of deep learning for predicting hemodynamics of human thoracic aorta. *J. Biomech.* **99**, 109544 (2020).
 21. Berg, P. *et al.* Multiple aneurysms anatomy challenge 2018 (MATCH): phase I: segmentation. *Cardiovasc. Eng. Technol.* **9**, 565–581 (2018).
 22. Radaelli, A. G. *et al.* Reproducibility of haemodynamical simulations in a subject-specific stented aneurysm model—a report on the Virtual Intracranial Stenting Challenge 2007. *J. Biomech.* **41**, 2069–2081 (2008).
 23. Varadarajan, A. V *et al.* Predicting optical coherence tomography-derived diabetic macular edema grades from fundus photographs using deep learning. *Nat. Commun.* **11**, 1–8 (2020).
 24. Jha, D. *et al.* Enhancing materials property prediction by leveraging computational and experimental data using deep transfer learning. *Nat. Commun.* **10**, 1–12 (2019).
 25. Guo, X., Li, W. & Iorio, F. Convolutional neural networks for steady flow approximation. in *Proceedings of the 22nd ACM SIGKDD international conference on knowledge discovery and data mining* 481–490 (2016).

REVIEWERS' COMMENTS:

Reviewer #1 (Remarks to the Author):

Thank you for taking the time to revise your manuscript. The paper reads well overall, but I still believe that, in lack of any biological/clinical validation, this work is more appropriate for a more technical journal (e.g. Nature Methods) instead of Nature Communications Biology that specifically “represents significant advances bringing new biological insight”, and this is not a direct outcome of this work.

Reviewer #2 (Remarks to the Author):

The authors have successfully addressed most of my previous concerns with the manuscript. In particular, they clarified the methods section to better describe architectural choices with respect to the original PointNet and introduced additional experiments to justify choices in the network architecture.

Furthermore, the discussion was extended with respect to the lacking evaluation on in-vivo data and claims were adjusted.

Two points remain open:

1.:

The discussion seems to be lacking a small paragraph on potential improvements when merging (parts of) all 4 networks.

Please comment on potential improvements due to similarities in features between the different fields and application scenarios.

2.:

"Global features were the global geometric information of the model. Local features referred to the distribution of each point and the corresponding flow field distribution inside the model. These two features both contained the geometric features of the same cardiovascular model, namely commonality. The two features also had different effective information, namely difference. The network needed to extract commonality and difference and learned the correlation between them to further realize the flow field prediction."

As commonality and difference are not per se known concepts but instead introduced to pick up in the next abstract I would propose to put them in brackets instead of "namely ...". I invite the authors to convince me of the opposite.

Minor issues:

Abstract:

20: In terms of sample resolution over 2 million of nodes...  ...of over 2 million nodes...

22: ...for applying complex arterial system...  ...for evaluating complex arterial systems...

32: Advanced deep learning algorithm...  Advanced deep learning algorithms...

39: Therefore, how to obtain hemodynamic...  Therefore, obtaining hemodynamic...

47: Even with ideal boundary conditions...  Even with simplified boundary conditions...

53: ..., CFD method also requires calculation time about ten minutes.  ..., CFD methods require a calculation time around ten minutes.

74: ...to adapt a flexibility and high resolution on the input geometry...  ...to adapt to the flexibility and high resolution of the input geometry...

85: In this study, we...  We...

106: ...was significantly reduced 600-fold.  ...was reduced 600-fold.

166: ...of the query point cloud could be obtained...  ...of the query point cloud be obtained...

215: ...was not sufficient enough to...  ...was not sufficient to...

277: Global features provide the outer geometry information of the model,...  Global features convey the outer geometry information within the model,...

Although the previous sentence states that the model itself computes the global features I propose to rephrase like this to avoid potential confusion.

292: ...that no matter whether the physical field has spatial components, the same network structure can achieve high accuracy prediction.  ...that the same network structure can achieve high accuracy predictions for physical fields with and without spatial components.

300: On the other hand, this assumption may also facilitate the stability of prediction by moderating the variety of hemodynamics.

Please rephrase this, potential instability when incorporating more realistic boundary conditions should not be put as a strength.

306: clinical  clinically

309: ...which is also one of the further work we need to accomplish.  ..., which we intend to address in the future.

331: ...can lead a variability of geometry.  ...can lead to variability in geometry.

332: ...of hemodynamic parameter...  ...of hemodynamic parameters...

372: ...on all wall boundary.  ...to all wall boundaries.

389: These four datasets need to be used as input to train the network independently. After that, we got four...  These four datasets were used independently to train four separate networks. Hence, we obtain four...

398: ...PointNet only had one...  ...PointNet had only one...

396: ...prediction of point cloud.  ...prediction of point clouds.

416: ...to the distribution of each point...  ...to the location of each point...

432: We stitched together the two vectors...  We concatenated the two vectors...

Supp Table 1: 32.8%<Error<1%  switch around numbers?

Supplement:

29: "Even the number of point cloud (nodes) varies, our network can accept that unfixed input. Actually, the 1100 models have different numbers of coronary arteries and different sizes of point clouds, but the network can still handle that. Combined with the universality analysis of the network, our deep learning method has many advantages."

I think this ending of the paragraph should be made a bit more concise.

RE: Manuscript ID COMMSBIO-20-1086A

Dear editor and dear reviewers,

We thank the editor for giving us the opportunity to submit a revised manuscript and the reviewers for their evaluations. We are grateful to all reviewers for their time and highly constructive comments. We try our best to address the concerns you raised in the form of our revised manuscript. All the changes we made are highlighted in the revised manuscript. This document lists the new parts point by point in the way of question and answer. We hope we could answer everything to your satisfaction and are looking forward to your feedback.

Best wishes,

Hitomi Anzai

Corresponding Author

Reviewer#1

Remarks to the Author: Thank you for taking the time to revise your manuscript. The paper reads well overall, but I still believe that, in lack of any biological/clinical validation, this work is more appropriate for a more technical journal (e.g. Nature Methods) instead of Nature Communications Biology that specifically “represents significant advances bringing new biological insight”, and this is not a direct outcome of this work.

Thank you very much for your comments. Based on your comments, we have made a related statement on the purpose and limitations of the manuscript. Due to the lack of clinical data, we are unable to carry out ‘clinical validation’ work. In the following work, we will further explore the possibility of improving this part.

Reviewer#2

Remarks to the Author: The authors have successfully addressed most of my previous concerns with the manuscript. In particular, they clarified the methods section to better describe architectural choices with respect to the original PointNet and introduced additional experiments to justify choices in the network architecture.

Furthermore, the discussion was extended with respect to the lacking evaluation on in-vivo data and claims were adjusted.

Thank you very much for your high evaluation and comments on our research. Based on your new comments, we have made the following amendments to the manuscript and explained in detail in the answers to specific questions below. Please check it.

Two points remain open:

1. **The discussion seems to be lacking a small paragraph on potential improvements when merging (parts of) all 4 networks.**

Please comment on potential improvements due to similarities in features between the different fields and application scenarios.

Answer:

Thank you very much for your comment. As suggested, we added a description of the existing limitations and potential improvements of this part in the Discussion(**Line295**): “We also noticed that the four data sets (preoperative, postoperative, velocity, and pressure fields) need to be trained separately as inputs, which increased the computational cost and operational complexity of deep learning to a certain extent. Therefore, we will explore potential improvements due to similarities in features between the different fields and application scenarios in further work. For example, by merging four data sets (with different labels), all prediction results can be output in one training session.”

2. **"Global features were the global geometric information of the model. Local features referred to the distribution of each point and the corresponding flow field distribution inside the model. These two features both contained the geometric features of the same cardiovascular model, namely commonality. The two features also had different effective information, namely difference. The network needed to extract commonality and difference and learned the correlation between them to further realize the flow field prediction."**

As commonality and difference are not per se known concepts but instead introduced to pick up in the next abstract I would propose to put them in brackets instead of "namely ...". I invite the authors to convince me of the opposite.

Answer:

Thank you very much for your comments, which helps us to make the logic of the paper correct and more rigorous. We fully agree with you that “As commonality and difference are not per se known concepts but instead introduced to pick up in the next abstract.” For a concept with a specific meaning proposed in this manuscript, it is inappropriate to use “namely” (applicable to a widely accepted, common-sense concept).

Based on your comments, we put them in brackets instead of “namely ...”. Please check it.

(Line421): “These two features both contained the geometric features of the same cardiovascular model (**commonality**). The two features also had different effective information (**difference**).”

Minor Issues:

- 1. 20: In terms of sample resolution over 2 million of nodes...  ...of over 2 million nodes...**

Answer:

We are very thankful for your comment. As suggested, we changed it in the revised manuscript: **(Line20)** “**In terms of over 2 million nodes...**” Please check it.

- 2. 22: ...for applying complex arterial system...  ...for evaluating complex arterial systems...**

Answer:

We are very thankful for your comment. As suggested, we changed it in the revised manuscript: **(Line22)** “**...for evaluating complex arterial system ...**” Please check it.

- 3. 32: Advanced deep learning algorithm...  Advanced deep learning algorithms...**

Answer:

We are very thankful for your comment. As suggested, we changed it in the revised sentence: **(Line62)** “**Advanced deep learning algorithms...**” Please check it.

- 4. 39: Therefore, how to obtain hemodynamic...  Therefore, obtaining hemodynamic...**

Answer:

We are very thankful for your comment. As suggested, we changed it in the revised sentence: **(Line39)** “**...obtaining hemodynamic ...**” Please check it.

- 5. 47: Even with ideal boundary conditions...  Even with simplified**

boundary conditions....

Answer:

We are very thankful for your comment. As suggested, we changed it in the revised sentence: **(Line52)** “Even with simplified boundary conditions...” Please check it.

- 6. 53: ..., CFD method also requires calculation time about ten minutes.  ..., CFD methods require a calculation time around ten minutes.**

Answer:

We are very thankful for your comment. As suggested, we changed it in the revised sentence: **(Line53)** “...CFD methods require a calculation time around ten minutes” Please check it.

- 7. 74: ...to adapt a flexibility and high resolution on the input geometry...  ...to adapt to the flexibility and high resolution of the input geometry...**

Answer:

We are very thankful for your comment. As suggested, we changed it in the revised sentence: **(Line75)** “...to adapt to the flexibility and high resolution of the input geometry...” Please check it.

- 8. 85: In this study, we...  We....**

Answer:

We are very thankful for your comment. As suggested, we changed it in the revised sentence: **(Line85)** “We...” Please check it.

- 9. 106: ...was significantly reduced 600-fold.  ...was reduced 600-fold.**

Answer:

We are very thankful for your comment. As suggested, we changed it in the revised sentence: **(Line106)** “...was reduced 600-fold”. We also removed other “Significant” from the manuscript, except for statistical tests. Please check it.

- 10. 166: ...of the query point cloud could be obtained...  ...of the query point cloud be obtained...**

Answer:

We are very thankful for your comment. In order to make sentences easier to read, we changed it in the revised sentence: **(Line165)** "...of the query point could be obtained..." Please check it.

- 11. 215: ...was not sufficient enough to...  ...was not sufficient to...**

Answer:

We are very thankful for your comment. As suggested, we changed it in the revised sentence: **(Line214)** "...was not sufficient to..." Please check it.

- 12. 277: Global features provide the outer geometry information of the model,...**
 Global features convey the outer geometry information within the model,...

Although the previous sentence states that the model itself computes the global features I propose to rephrase like this to avoid potential confusion..

Answer:

We are very thankful for your comment. As suggested, we changed it in the revised sentence: **(Line276)** "Global features convey the outer geometry information within the model". Please check it.

- 13. 292: ...that no matter whether the physical field has spatial components, the same network structure can achieve high accuracy prediction.  ...that the same network structure can achieve high accuracy predictions for physical fields with and without spatial components.**

Answer:

We are very thankful for your comment. As suggested, we changed it in the revised sentence: **(Line291)** "...that the same network structure can achieve high accuracy predictions for physical fields with and without spatial components." Please check it.

- 14. 300: On the other hand, this assumption may also facilitate the stability of prediction by moderating the variety of hemodynamics.**
Please rephrase this, potential instability when incorporating more realistic boundary conditions should not be put as a strength.

Answer:

We are very thankful for your comment. We fully agree with you. Based on your comment, we delete this unsuitable sentence here. Please check it.

(Line305) “Currently we adopted constant values on inlets and outlets, which have been widely used among a number of geometries⁴⁴⁻⁴⁶. Therefore, the simulation results should include differences from real hemodynamics. In future approaches that include boundary conditions, another input channel will be required on the network. This input channel will take the patient's personalized boundary conditions as the input, and together with the model point cloud as the teaching signal to control the training process.”

15. 306: clinical  clinically.

Answer:

We are very thankful for your comment. As suggested, we changed it in the revised sentence: **(Line310)** “clinically”. Please check it.

16. 309: ...which is also one of the further work we need to accomplish.  ..., which we intend to address in the future.

Answer:

We are very thankful for your comment. As suggested, we changed it in the revised sentence: **(Line313)** “...which we intend to address in the future”. Please check it.

17. 331: ...can lead a variability of geometry.  ...can lead to variability in geometry.

Answer:

We are very thankful for your comment. As suggested, we changed it in the revised sentence: **(Line334)** “...can lead to variability in geometry”. Please check it.

18. 332: ...of hemodynamic parameter...  ...of hemodynamic parameters...

Answer:

We are very thankful for your comment. As suggested, we changed it in the revised sentence: **(Line336)** “...of hemodynamic parameters...” Please check it.

19. 372: ...on all wall boundary.  ...to all wall boundaries.

Answer:

We are very thankful for your comment. As suggested, we changed it in the revised sentence: **(Line375)** "...to all wall boundaries." Please check it.

- 20. 389: These four datasets need to be used as input to train the network independently. After that, we got four...  These four datasets were used independently to train four separate networks. Hence, we obtain four...**

Answer:

We are very thankful for your comment. As suggested, we changed it in the revised sentence: **(Line392)** "These four datasets were used independently to train four separate networks. Hence, we obtain four..." Please check it.

- 21. 398: ...PointNet only had one...  ...PointNet had only one...**

Answer:

We are very thankful for your comment. As suggested, we changed it in the revised sentence: **(Line401)** "PointNet had only one..." Please check it.

- 22. 396: ...prediction of point cloud.  ...prediction of point clouds.**

Answer:

We are very thankful for your comment. As suggested, we changed it in the revised sentence: **(Line399)** "...prediction of point clouds" Please check it.

- 23. 416: ...to the distribution of each point...  ...to the location of each point...**

Answer:

We are very thankful for your comment. As suggested, we changed it in the revised sentence: **(Line419)** "...to the location of each point..." Please check it.

- 24. 432: We stitched together the two vectors...  We concatenated the two vectors...**

Answer:

We are very thankful for your comment. As suggested, we changed it in the revised sentence: **(Line435)** "We concatenated the two vectors..." Please check it.

25. Supp Table 1: 32.8%<Error<1%  switch around numbers?

Answer:

We are very thankful for your comment. What you understand is correct. Due to different parameter settings (e.g. Reynolds number, etc.), Lee's predicted ERR values had a big difference (“switch around numbers”).

26. Supplement:

29: "Even the number of point cloud (nodes) varies, our network can accept that unfixed input. Actually, the 1100 models have different numbers of coronary arteries and different sizes of point clouds, but the network can still handle that. Combined with the universality analysis of the network, our deep learning method has many advantages."

I think this ending of the paragraph should be made a bit more concise.

Answer:

We are very thankful for your comment. It makes our sentences more concise. According to the suggestion, we deleted the repetitive part: **(Supplement Information Line29)** “Even the number of point cloud (nodes) varies, our network can accept that unfixed input. Combined with the universality analysis of the network, our deep learning method has many advantages.” Please check it.